# Iterative Compositional Data Generation for Robot Control

**Anh-Quan Pham**[1][*]   **Marcel Hussing**[1]   **Shubhankar P. Patankar**[1]
**Dani S. Bassett**[1]   **Jorge Mendez-Mendez**[2]   **Eric Eaton**[1]

[1]*University of Pennsylvania*   [2]*Stony Brook University*

Reviewed on OpenReview: `https://openreview.net/forum?id=cASorO1kiy`

## Abstract

Collecting robotic manipulation data is expensive, making it impractical to acquire demonstrations for the combinatorially large space of tasks that arise in multi-object, multi-robot, and multi-environment settings. While recent generative models can synthesize useful data for individual tasks, they do not exploit the compositional structure of robotic domains and struggle to generalize to unseen task combinations. We propose a semantic compositional diffusion transformer that factorizes transitions into robot-, object-, obstacle-, and objective-specific components and learns their interactions through attention. Once trained on a limited subset of tasks, we show that our model can zero-shot generate high-quality transitions from which we can learn control policies for unseen task combinations. Then, we introduce an iterative self-improvement procedure in which synthetic data is validated via offline reinforcement learning and incorporated into subsequent training rounds. Our approach substantially improves zero-shot performance over monolithic and hard-coded compositional baselines, ultimately solving nearly all held-out tasks and demonstrating the emergence of meaningful compositional structure in the learned representations.

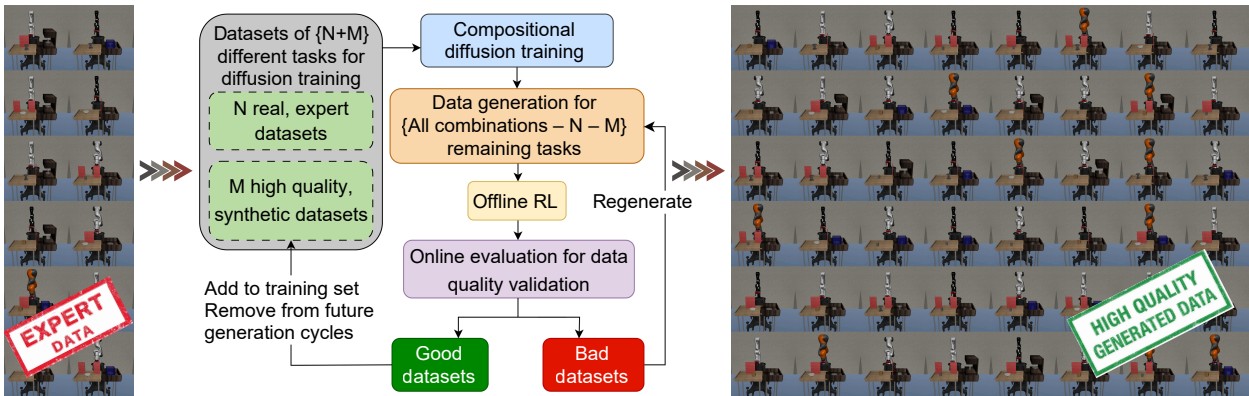

Figure 1: Iterative Compositional Data Generation.

## 1   Introduction

Augmenting model training with self-generated data is a promising approach to improve sample efficiency in domains where collecting real data is expensive. In the context of robotic manipulation, gathering new

---

[*]Corresponding author: `quanpham@seas.upenn.edu`
Project website (visualizations and code for reproducibility): `anhquanpham.github.io/iterative-comp-rl-generation`

experience requires operating a robot—a process that is not only labor- and time-intensive, but also demands expert knowledge, either to collect datasets or to monitor reinforcement learning (RL) training and scale policies to obtain datasets across a wide range of task variations. Consequently, collecting data from scratch for every possible new manipulation task quickly becomes impractical as evidenced by various large-scale data collection efforts (Walke et al., 2023; O'Neill et al., 2024; Khazatsky et al., 2024). Recent work has shown that current generative models can produce data of sufficient quality to enable training models with substantially reduced real experience, including in control settings (Yu et al., 2023; Lu et al., 2023; Liang et al., 2023). However, most existing approaches focus on improving sample efficiency within a single task, and do not leverage self-generated data to accelerate learning on entirely new tasks (Janner et al., 2022; Lu et al., 2023).

In this work, we investigate whether a robot learning system can iteratively improve its ability to solve unseen tasks by generating artificial training data for those unseen tasks with a self-improving generative model (Figure 1). We leverage the insight that cross-embodiment robotic manipulation domains exhibit an inherent compositional structure, whereby each task solution involves a unique composition of reusable models of objects, skills, and controllers. Our central hypothesis is that constructing model architectures that explicitly exploit this compositionality enables *zero-shot generation of high-quality synthetic training data* for novel task compositions, mitigating the need to relearn or collect data for every task from scratch.

We focus on a reinforcement learning (RL) setting in which tasks are defined compositionally (Mendez et al., 2022a; Hussing et al., 2024), constructed by combining a small number of elements such as robots, objects, obstacles, and goals. Intuitively, machine learning approaches can exploit the inherent combinatorial structure of these domains to generalize to unseen task configurations. Yet, standard single-and multi-task agents require vast amounts of data in such settings, struggling to exploit the compositional structure when the available data is small. Learners are better able to exploit the structure when the policy architecture mirrors the underlying task factorization (Devin et al., 2017; Andreas et al., 2017; Mendez et al., 2022a;b).

One outstanding challenge is that pre-defining such architectures requires strong prior knowledge about the correct decomposition. While much prior engineering knowledge is available for the robotics tasks we consider, it is unclear that these priors are optimal. In this work, we instead train a transformer to learn the compositional structure directly from data, leveraging the transformer interpretation as a graph neural network (GNN). Instead of training a policy on a subset of task combinations and evaluating its zero-shot capabilities, we train a diffusion transformer on the same subset of tasks to *generate training data* for the policies of unseen task combinations, thereby reducing the amount of data required to learn novel behaviors. The model learns a separate tokenizer for each individual task module (e.g., a specific robot, object, or environment) and uses cross-attention to infer the graph that connects these encoders. This yields a representation that is analogous to the hard-coded compositional network used in earlier work (Mendez et al., 2022a), but the structure is learned from data rather than specified a priori.

We first demonstrate that our task-conditional diffusion transformer enables superior zero-shot generalization capabilities compared to monolithic architectures. We then highlight the need for compositional tokenization by showing that models with factor-specific tokenizers achieve improved zero-shot performance relative to models that rely on a single shared tokenizer. Next, we show that models that properly learn the underlying task decomposition can be iteratively trained on their own generated data for unseen tasks to produce training data for solving *more* new tasks without requiring additional real data. We further analyze the learned representations and find that the model discovers a decomposition that differs from previous work, indicating that effective compositional structure can emerge automatically from data. Finally, we provide additional empirical results demonstrating the efficiency of our approach, the utility of synthetic datasets even when successful trajectories are rare, and the scalability of the method to larger compositional task spaces.

## 2 Preliminaries

We formulate our problem as generating data of and learning policies in a Markov decision process (MDP) $\mathcal{M}_n = (S_n, A_n, R_n, \mathcal{P}_n, T)$ where $S_n$ is the state space, $A_n$ is the action space, $R_n$ is the reward function mapping state-action pairs $(s, a)$ to a scalar $r$, $\mathcal{P}_n$ is the transition probability function determining the next

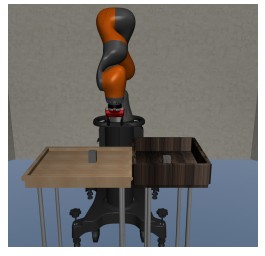 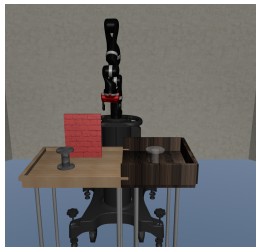 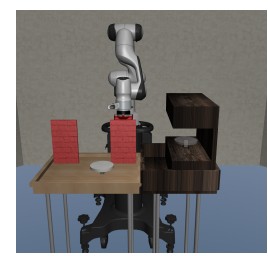 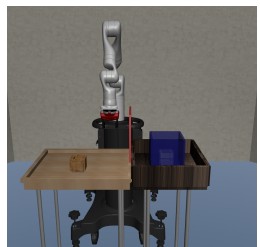

(a) `IIWA`, `Box`, `None`, `PickPlace`

(b) `Jaco`, `Dumbbell`, `ObjectWall`, `Push`

(c) `Panda`, `Plate`, `ObjectDoor`, `Shelf`

(d) `Kinova3`, `Hollowbox`, `GoalWall`, `Trashcan`

Figure 2: Four example CompoSuite tasks, each defined by selecting one element from each axis.

| Robot ID | | | | Object ID | | | | Obstacle ID | | | | Subtask ID | | | |
|---|---|---|---|---|---|---|---|---|---|---|---|---|---|---|---|
| IIWA | Jaco | Panda | Kinova3 | Box | Dumbbell | Plate | Hollow box | None | Object Wall | Object Door | Goal Wall | Pick Place | Push | Shelf | Trash can |

Kinova3 Plate None Push ➜ [0, 0, 0, 1, 0, 0, 1, 0, 1, 0, 0, 0, 0, 1, 0, 0]

Figure 3: Overview of the 16-dimensional task indicator. For every task in CompoSuite, the model receives a binary vector formed by concatenating four one-hot segments: a 4-dimensional robot ID, 4-dimensional object ID, 4-dimensional obstacle ID, and 4-dimensional subtask ID. Each segment activates exactly one entry corresponding to the chosen element along that axis. The example demonstrates how the task `Kinova3`, `Plate`, `None`, `Push` is encoded into the final 16-element vector.

state $s'$ given the current state-action pair $(s, a)$, and $T$ is the task horizon. The rewards are bounded to the range $r \in [0, 1]$. We say a agent succeeds if, at any state in a trajectory, it achieves the maximum reward of $r = 1$. We also define the function $D_n : S_n \mapsto A_n$ which outputs a termination signal $d$ that indicates if the agent has moved to an absorbing, non-rewarding state. We consider a set of $N$ such MDPs $\{\mathcal{M}_n\}_{n=0}^{N-1}$ and our goal is to learn a policy $\pi^* = \{\pi_n^*\}_{n=0}^{N-1}$ that maximizes the average probability of success over this set, that is, $\pi^* \in \arg\max_\pi \frac{1}{N} \sum_{n=0}^{N-1} \mathbb{E}_{\pi_n, \mathcal{M}_n} \big[ \mathbf{1}[\max_{0 \leq t \leq T} r_t = 1] \big]$.

## 2.1 CompoSuite Benchmark

CompoSuite (Mendez et al., 2022a) is a simulated robotic manipulation benchmark for evaluating compositional reinforcement learning (RL) agents. CompoSuite provides a family of $4 \times 4 \times 4 \times 4 = 256$ distinct manipulation tasks by composing exactly one out of four elements from each of the following four axes:

- **Robot arms**  KUKA's IIWA, Kinova's Jaco, Franka's Panda, Kinova's Gen3.
- **Objects**  Box, Dumbbell, Plate, Hollow box.
- **Obstacles**  No obstacle, Wall blocking object, Doorway near object, Wall blocking goal.
- **Objectives**  Pick and place, Push, Place on shelf, Place in trash can.

For each task, states are provided as symbolic representations containing proprioceptive robot features (joint and gripper positions and velocities) together with absolute and relative Cartesian positions of the object, obstacle, and goal in the scene. Each task is defined by selecting exactly one unique element from each of the four axes: Robot, Object, Obstacle, and Objective. To illustrate the compositional structure, Figure 2 shows four example tasks from CompoSuite. The state vector also contains a binary indicator vector of length 16 that identifies a task via four one-hot sub-vectors, one for each axis. Figure 3 illustrates this layout. Rewards are defined with dense, stage-wise rewards to guide the learning.

Hussing et al. (2024) released 1 million transitions for every task in CompoSuite across four dataset variants (approximately 1 billion transitions in total). The four datasets span a range of performance levels, from early training to expert proficiency, and were generated using policies trained with Proximal Policy Opti-

mization (Schulman et al., 2017) and Soft Actor-Critic (Haarnoja et al., 2018). Trajectories are stored as transition tuples $\langle s, a, r, s', d \rangle$. For our experiments, we focus exclusively on the expert datasets.

## 2.2 Diffusion Models

A diffusion model is a generative model that learns to reverse a gradual noising process applied to data (Ho et al., 2020). We use diffusion models to generate artificial data for training. More precisely, we use the Elucidated Diffusion framework (Karras et al., 2022). Given a data vector $x_0 \in \mathbb{R}^d$, we consider a collection of noise levels $\{\sigma_t\}_{t=1}^T$ with $\sigma_t > 0$. For each $t$, the forward process $q$ produces a noised sample $x_t$ by adding Gaussian noise of magnitude $\sigma_t$:

$$x_t = x_0 + \sigma_t \varepsilon \ , \quad \varepsilon \sim \mathcal{N}(0, I) \ , \quad \log \sigma_t \sim \mathcal{N}(P_{\text{mean}}, P_{\text{std}}^2) \ , \quad t = 1, \ldots, T \ ,$$

so that, for any fixed noise level $\sigma_t$, the conditional distribution of $x_t$ given $x_0$ is $q(x_t \mid x_0, \sigma_t) = \mathcal{N}(x_0, \sigma_t^2 I)$. Here, $P_{\text{mean}} \in \mathbb{R}$ and $P_{\text{std}} > 0$ are scalar hyperparameters that control the mean and standard deviation of the log-noise distribution $\log \sigma_t$. A neural network $\varepsilon_\theta$ is trained to predict the clean sample $x_0$ from a noised sample $x_t$ and its associated noise level $\sigma_t$. The training objective is a noise-weighted reconstruction loss:

$$\mathcal{L}_{\text{diff}}(\theta) = \mathbb{E}_{x_0 \sim p_{\text{data}}, \sigma_t, \varepsilon \sim \mathcal{N}(0, I)} \left[ \| \varepsilon_\theta(x_t, \sigma_t) - x_0 \|_2^2 \, w(\sigma_t) \right] \ , \quad w(\sigma) = \frac{\sigma^2 + \sigma_{\text{data}}^2}{(\sigma \, \sigma_{\text{data}})^2} \ ,$$

where $\sigma_{\text{data}} > 0$ is a hyperparameter representing the typical data scale. At generation time, the model constructs a reverse denoising process over a decreasing sequence of noise levels $\{\sigma_t\}_{t=1}^T$. Starting from a high-noise initialization $x_T \sim \mathcal{N}(0, \sigma_T^2 I)$, the model iteratively applies the denoiser to define reverse transitions $p_\theta(x_{t-1} \mid x_t)$ until it obtains a synthetic sample $x_0$ that approximately follows the data distribution.

## 3 Task-Graph Compositional Transformer for Iterative Data Generation

We assume a *functionally compositional* task graph. Mendez et al. (2022b) define a hard-coded set of modules, each representing a task element, and define task solutions as fixed paths through that graph. We instead assume that each task consists of basic elements, each of which is a random variable corresponding to one component of the transition, such as a factor of the state or next state, an action, a reward, or a termination indicator. Let $\mathcal{F}$ denote the set of all such basic elements. Then, each element $f \in \mathcal{F}$ is associated with an input space $X^f$ and a representation space $Y^f$. An encoder-decoder pair $(e_f, o_f)$ maps raw variables into the representation space $e_f : X^f \mapsto Y^f$ and back $o_f : Y^f \mapsto X^f$. We define a computation graph $G = (V, E)$ that captures the shared structure across all tasks, where the vertices $V = Y^f_{f \in \mathcal{F}}$ are the representation spaces and the edges $E$ are transformations that specify how information can flow between representations. A specific MDP $\mathcal{M}_n$ is characterized by a subset of elements present in that task, $\mathcal{F}_n \subseteq \mathcal{F}$, and the induced subgraph of $G$ on their representation spaces, with a joint distribution over their values. The CompoSuite benchmark fits this view: a task is obtained by selecting one element along the robot, object, obstacle, and objective axes, thereby instantiating a particular set of state-factor vertices and their interactions.

As the graph operates in representation space, it can be used to instantiate a variety of learned functions on an MDP, such as a policy or a conditional generative model. A probabilistic model defined on $G$ can specify a conditional distribution over the unobserved basic elements given the values of any subset of observed ones.

## 3.1 Transformers as Graphs

Hard-coding the structure of the computation graph requires extensive domain knowledge and may result in a suboptimal architecture. In consequence, we would like to learn the graph structure directly from data. For this, we rely on the finding that the well-known transformer architecture (Vaswani et al., 2017) can be interpreted as a GNN (Joshi, 2025). In particular, a transformer with $L$ layers maps a sequence of input tokens $x_1, \ldots, x_K$ to a sequence of output representations $h_1^L, \ldots, h_K^L$ by repeatedly applying self-attention and feed-forward layers. For each token $i$ and layer $\ell$, the model computes queries, keys, and values $q_i^\ell = W_Q h_i^{\ell-1}$, $k_j^\ell = W_K h_j^{\ell-1}$, $v_j^\ell = W_V h_j^{\ell-1}$ where $W_Q, W_K, W_V$ are learned weight matrices. The

model then assigns the $i$-th token's attention weights over each other token $j$ as $\alpha_{ij}^\ell = \mathrm{softmax}j\left(q_i^{\ell\top} k_j^\ell / \sqrt{d}\right)$ and aggregates other tokens' values into an updated representation $h_i^\ell = \mathrm{FF}\left(\sum j = 1^N \alpha_{ij}^\ell v_j^\ell\right)$, where FF implements a feed-forward layer and $h_i^0 = x_i$. In the interpretation of the transformer as a GNN, each token $i$ corresponds to a node with a feature vector $h_i^{\ell-1} \in \mathbb{R}_h^d$, and self-attention implements message passing on a fully connected directed graph over these nodes. By learning the weight matrices $W_Q, W_K, W_V$, the transformer learns to assign high attention from token $i$ to token $j$ exactly when the graph should contain a strong directed edge from node $j$ to node $i$.

Interpreting the transformer as a GNN suggests that a transformer can automatically discover the underlying graph structure of a set of problems that are related compositionally. This reduces the architectural challenge of designing an appropriate graph to designing a tokenization scheme that enables representing such a graph.

## 3.2 Semantic Compositional Diffusion Transformers

Thus, we set out to encode our task graph in a diffusion model by implementing $\epsilon_\theta$ as a diffusion transformer (DiT; Peebles & Xie, 2023). This model processes noised inputs $(x_{t,1}, \ldots, x_{t,K})$ in the original transition space and outputs denoised predictions $\epsilon_\theta(x_t, t) \in \mathbb{R}^{K \times d_x}$ at each diffusion step, which is interpreted as a prediction of the added noise for each component. Our diffusion transformer architecture internally uses factor-specific encoders to map inputs to token embeddings, processes these through self-attention, and decodes back to the original space at each step of the reverse diffusion process.

We construct the input sequence for each transition by associating each component of our task graph $f \in \mathcal{F}$ with the elements of CompoSuite as described in Section 2.1. For both the state and next state, we treat each element per axis as one factor—e.g. each robot arm is one factor. In addition, we add one factor each for the action, reward, and termination signal. This yields a DiT that can learn directly on top of the task graph. For representation learning, we equip each factor with a parametric encoder-decoder pair $(e_{f,\theta}, o_{f,\psi})$, both instantiated as neural networks. The encoder maps the inputs into a learned embedding $y^f = e_{f,\theta}(z^f)$, which we interpret as living in the representation space $Y^f$. The collection $(y^f)_{f \in \mathcal{F}}$ is treated as the $K$ tokens processed by the transformer. Self-attention over this factor-specific set of tokens implements graph-compositional inference: at each diffusion step, the representation of every factor $f$ is updated by attending to all other factors $f' \in \mathcal{F}$, so that, for example, the current robot token can condition on the current object, obstacle, and objective tokens in a way that mirrors the edges of $G = (V, E)$. At each diffusion step, the transformer outputs denoised token embeddings $\bar{y}^f$, which are then mapped back to the original variable domains with the decoders, yielding predictions $(o f, \psi(\bar{y}^f)) f \in \mathcal{F}$ in the original transition space.

For conditioning, the original DiT injects variables such as the diffusion step $t$ through adaptive layer normalization. This produces per-block scale and shift parameters that gate the self attention and feed-forward updates. We implement our task conditioning via an additional input embedding that modulates all transformer blocks. For each diffusion step $t$ and task index $n$, we form a context embedding $u(t, n) = E_t(t) + E_n(n)$ and pass it through a small network that produces adaptive normalization and gating parameters for every block of the DiT. This pathway injects $(t, n)$ into the model only through these adaptive transforms, and leaves the compositional semantics of the factor-specific tokenization unchanged. The resulting network provides us with a diffusion model that can be trained to generate RL transitions for each task by simply selecting the correct encoders and conditioning. We visualize our proposed architecture in Figure 4.

Using this architecture for diffusion modeling induces a joint representation over factor-specific component embeddings. For each task index $n$, the learned diffusion model defines a distribution $p_\theta(x_0 \mid n)$ over denoised transitions $x_0$ in the original transition space. Within the denoiser, factor-specific encoders map each component to token embeddings $(\bar{y}^f) f \in \mathcal{F}$, the shared diffusion transformer blocks (modulated by task conditioning through adaptive layer normalization) process these through self-attention to learn relationships between factors, and decoders map the embeddings back to the original space. As this joint representation over component embeddings is shared across tasks, we can use structure learned from one task to improve the marginals for others and incrementally refine each factor's predictive distribution as new tasks are added.

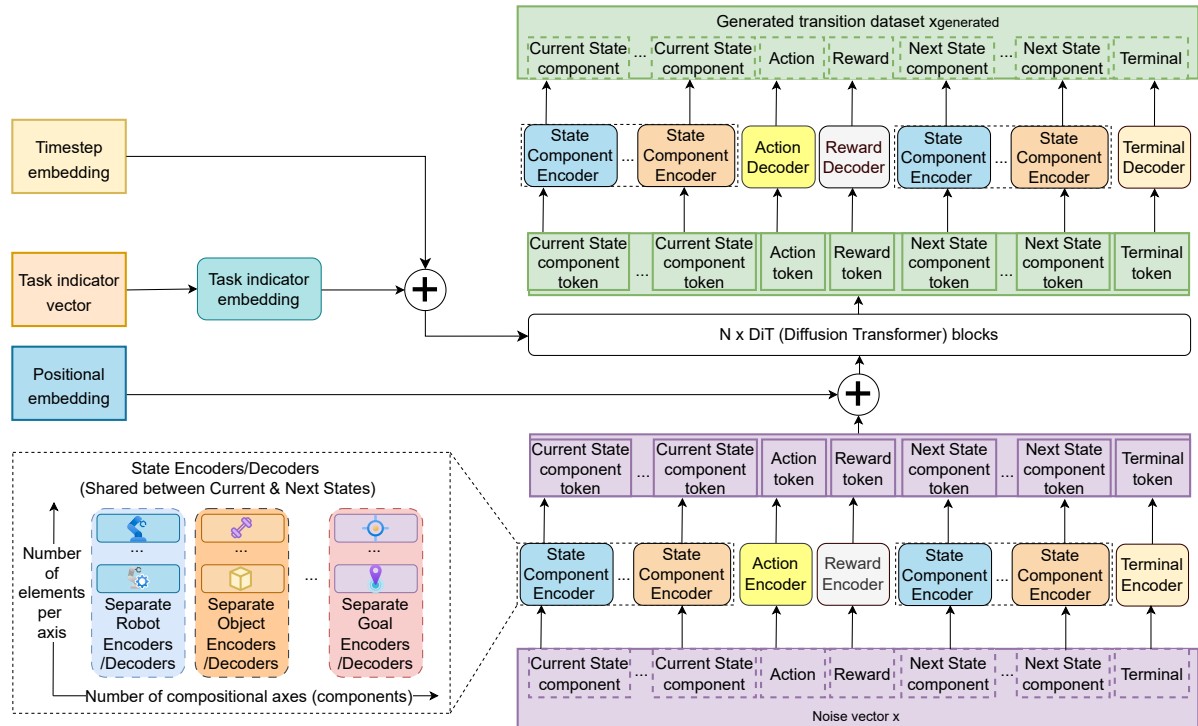

Figure 4: Visualization of our semantic compositional transformer architecture. We factorize each transition into state factors, actions, reward, and terminal indicators. State factors represent the compositional dimensions along which tasks can vary (for example, different robots). Each state factor has its own encoder-decoder pair, providing partial parameter separation. The encoded tokens, augmented with positional embeddings, are processed by several diffusion transformer layers. The diffusion transformer uses adaptive statistics conditioned on timestep and task indicator embeddings. Its output tokens are then decoded by the factor-specific decoders, with state encoder-decoder pairs shared between current and next state.

## 3.3 Self-Refining Compositional Distributions

We use our trained DiT to produce training data for two purposes: training behavior policies for unseen tasks (for which we do not have real training data), and updating the DiT itself. By virtue of the compositional graph structure, our model can train on a set of compositional tasks, generate data for new task combinations zero-shot, and use the generated data to improve the DiT. Notably, because we learn factorized pieces of a distribution to improve their marginals, the generator improves not only on the tasks for which we generate data, but also on tasks that share some of these factors. As an illustrative example, consider tasks that vary on the robot and object (e.g., a subset of CompoSuite). Suppose the dataset contains transitions for three tasks: (Panda, Box), (Jaco, Box), and (Panda, Plate), but no real data for (Jaco, Plate). Our compositional DiT can generate plausible transitions for (Jaco, Plate) by combining the learned Jaco and Plate factors. Retraining on these additional samples increases the effective data available for the shared Plate factor and constrains it across multiple robotic contexts, so that any downstream task involving Plate —e.g., (IIWA, Plate)—benefits from a sharper object marginal than would be possible from the original tasks alone.

We summarize our procedure in Algorithm 1. The algorithm starts with a set of (real) data from a sub-set of training tasks and proceeds in rounds. In every round, we fit our DiT to all training data. We then generate data for all existing validation and test tasks, train a policy using TD3-BC (Fujimoto & Gu, 2021) on the generated data, and evaluate the policy online in the environment. If the success rate (sr) for any task is larger than some threshold $\tau$, we add the data generated for this task to the training set. This performance-based filter only admits synthetic data that is useful for training high-quality policies. If no tasks surpass

---

**Algorithm 1** Compositional Iterative Bootstrapping with Synthetic Data

---

**Require:** Initial dataset $\mathcal{D}$; target task set $\mathcal{T}_{\text{target}}$; initial threshold $\tau_0$ with lower bound $\tau_{\min}$; threshold step size $\Delta_\tau$; patience $C$; maximum iterations $K$

1: $\mathcal{D}_{\text{train}}^{(0)} \leftarrow \mathcal{D}$, $\mathcal{T}_{\text{solved}}^{(0)} \leftarrow \emptyset$, $\tau \leftarrow \tau_0$, $c \leftarrow 0$
2: **for** $k = 0, 1, \ldots, K$ **do**
3:      Train diffusion model $\varepsilon_\theta^{(k)}$ on $\mathcal{D}_{\text{train}}^{(k)}$
4:      **for all** $t \in \mathcal{T}_{\text{target}} \setminus \mathcal{T}_{\text{solved}}^{(k)}$ **do**
5:          Generate synthetic data $\mathcal{D}_{\text{syn}}^{(k)}(t)$ using $\varepsilon_\theta^{(k)}$
6:          Train and evaluate TD3-BC policy $\pi_\phi^{(k)}(t)$ on $\mathcal{D}_{\text{syn}}^{(k)}(t)$ to obtain success rate $\text{sr}^{(k)}(t)$
7:      **end for**
8:      $\mathcal{T}_{\text{solved}}^{(k+1)} \leftarrow \mathcal{T}_{\text{solved}}^{(k)} \cup \{t : \text{sr}^{(k)}(t) > \tau\}$
9:      **if** $\mathcal{T}_{\text{solved}}^{(k+1)} = \mathcal{T}_{\text{solved}}^{(k)}$ **then** $c \leftarrow c + 1$ **else** $c \leftarrow 0$ **end if**
10:     **if** $c \geq C$ **then** $\tau \leftarrow \max(\tau - \Delta_\tau, \tau_{\min})$, $c \leftarrow 0$ **end if**
11:     $\mathcal{D}_{\text{train}}^{(k+1)} \leftarrow \mathcal{D} \cup \bigcup_{t \in \mathcal{T}_{\text{solved}}^{(k+1)}} \mathcal{D}_{\text{syn}}^{(k)}(t)$
12:     **if** $\mathcal{T}_{\text{target}} \subseteq \mathcal{T}_{\text{solved}}^{(k+1)}$ **then break end if**
13: **end for**
14: **return** $\varepsilon_\theta^{(k)}, \{\pi_\phi^{(k)}(t)\}_{t \in \mathcal{T}_{\text{target}}}, \mathcal{T}_{\text{solved}}^{(k)}, \mathcal{D}_{\text{train}}^{(k)}$

---

the quality threshold, we increase a patience parameter $c$. When the patience exceeds a predefined threshold $C$, we decrease the quality threshold $\tau$.

The approach in Algorithm 1 uses self-generated data to train a data generator. One question is whether adding sub-optimal data for a single task might lead to degraded performance across all tasks. Our compositional transformer architecture from Section 3.2 uses the data generated for a particular task exclusively to train the encoder-decoder pairs specific to that task's elements. In consequence, each generated dataset contributes only to a subset of all transformer weights (e.g., data generated for the `Panda` encoder-decoder is not used to update the parameters of the `Jaco` encoder-decoder). This modular structure localizes task-specific updates to the corresponding encoder–decoder modules, which can limit the extent to which low-quality data from one task affects representations used by others.

## 4 Experimental Evaluation of the Semantic Compositional Transformer

This section empirically evaluates our compositional transformer for generating robotic data of unseen tasks. Because our algorithm generates millions of synthetic transition tuples for each task, we restrict our experiments to a subset of 64 tasks of the CompoSuite benchmark, chosen by using one fixed robot arm: IIWA. We consider a setting where we only have training data for 14 of the 64 tasks, which we show empirically to be insufficient for zero-shot generalization of a non-compositional data-generating baseline (Appendix 4.7.1). We use our method to iteratively generate data for the remaining 50 tasks, and report performance on a test set consisting of 32 of the 50 held-out tasks. Note that Algorithm 1 requires online evaluation on the zero-shot tasks, for which we perform 10 trajectory rollouts per task (500 transitions per rollout) every round.

### 4.1 Baselines

We first compare against two static offline RL approaches, which cannot generate data for new tasks, to demonstrate the value of iterative data generation.

- **Hardcoded Compositional RL** We train the multi-task compositional architecture of Mendez et al. (2022a) via offline RL using TD3-BC. This architecture was designed specifically to solve CompoSuite tasks, but it employs a hard-coded compositional structure.
- **Semantic Compositional RL** To test the benefits of learned connections in compositional representations, we also implement a semantic compositional RL approach based on our architecture. We use TD3-BC to train a multi-task model which uses our semantic compositional transformer for the encoder.

However, rather than decoding each element with its own decoder, we use mean pooling over all output tokens and process the concatenated vector with an additional feed-forward layer to obtain an action.

We then consider three baseline architectures that iteratively generate data per Algorithm 1. To disentangle improvements due to compositional structure from architectural choices, we compare diffusion models including a monolithic diffusion model following SynthER (Lu et al., 2023) with a feed-forward denoiser, as well as transformer-based diffusion models that differ in their tokenization and compositional structure.

- **Monolithic** To highlight the difficulty of compositional generalization for monolithic architectures, we consider a variant of Synthetic Experience Replay (SynthER; Lu et al., 2023). SynthER trains a diffusion model on the data collected by an off-policy RL algorithm (e.g., TD3) to augment the RL batch with artificial transitions. We are specifically interested in the neural network architecture for diffusion, as it has shown promise for generating useful transitions for RL training. In particular, SynthER employs a monolithic architecture that parametrizes the diffusion denoiser $\epsilon_\theta$ via several residual feed-forward layers. We adapt this architecture to the multi-task setting by conditioning the denoiser $\epsilon_\theta$ on the task indicator. At each layer, the noisy transition is augmented by additive embeddings encoding both timestep and task information $\widetilde{x}_t = x_t + E_t(t) + E_c(c)$, where $E_t(t)$ encodes the diffusion timestep through sinusoidal features and $E_c(c)$ linearly projects the multi-hot task indicator into the same latent space. This conditioning strategy injects task information into the residual computation without modifying the architecture.
- **Standard DiT** We then compare against a standard DiT without semantic or compositional tokenization (Peebles & Xie, 2023). This DiT simply chops the input into patches of roughly size 15 and computes the tokens using a shared encoder. This yields a transformer with the same amount of tokens as our compositional semantic encoder but without compositional structure.
- **Semantic DiT** Our tokenization scheme splits the input into semantic patches (e.g., robot state, object state, action) and uses a separate encoder-decoder pair for each element (e.g., one for the IIWA and one for the Jaco). To verify the need to train separate encoder-decoders to learn the different representation spaces for each element, we compare against a DiT that splits the input into semantic patches but trains one shared encoder-decoder across elements of an axis (e.g., one for all robots). While this carries the semantic meaning of the input, it does not model the nodes that constitute the CompoSuite task graph.

In each round of data generation, the diffusion model generates data for the held-out tasks that have not surpassed the threshold $\tau$ (i.e., unseen tasks for the diffusion model). We use the generated data to train task-specific RL policies using TD3-BC for 50,000 steps, rolling out 10 evaluation trajectories every 5,000 steps. We keep the best-performing policy for a task across evaluation steps and data generation iterations.

## 4.2 Zero-shot Generalization

To verify the zero-shot abilities of our approach, we pre-train all models on the training tasks and report success on the held-out test tasks. RL baselines directly use a zero-shot policy, while diffusion approaches generate synthetic data and train policies on the generated data. We also run four iterations of iterative self-improvement (Algorithm 1) on diffusion approaches. Performance of RL algorithms is averaged across tasks over 15 seeds. For generative models, we keep the best-performing policy across the four iterations. We train the diffusion model over three independent seeds, and for each seed we train the policies over five RL training seeds; we report the average across tasks, diffusion seeds, and RL seeds. Error bars indicate standard error across 15 RL seeds and three diffusion seeds. We report the results in Figure 5.

**Reinforcement learning performance** The compositional RL baseline of Mendez et al. (2022a), which was specifically designed for these tasks, achieves some zero-shot generalization. However, the composition learned by our compositional transformer succeeds twice as often. This provides evidence that our architecture can extract meaningful compositional structure from data. The improved performance suggests that the graph structure learned by our architecture more effectively connects relevant vertices than the hard-coded architecture of Mendez et al. (2022a).

**Initial generative performance** After the first round of training the first diffusion models, RL based on the data generated by the monolithic architecture performs worse than all compositional variants (RL or diffusion), demonstrating the usefulness of composition for efficient zero-shot data generation. The standard DiT model performs worst across all models, indicating that a proper tokenization of the input space is

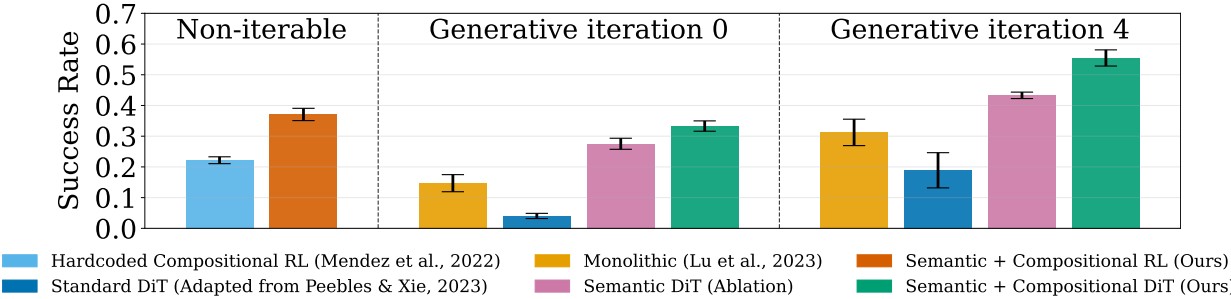

Figure 5: Zero-shot success rate for non-iterable RL models and RL models trained on synthetic data of a generative model at iteration iteration 0, and best zero-shot success rate for RL models trained on synthetic of a generative model after 4 iterations of self-improvement. Overall, the semantic compositional architecture leads to large improvements in the RL as well as generative model setting. (Left) The semantic compositional architecture improves over the hard-coded compositional RL architecture by almost doubling the success rate. (Middle) In the generative model setting, the semantic tokenization approaches beat the architectures that are not adapted to the task. (Right) After 4 rounds of self-improvement, the semantic compositional DiT method achieves highest performance across all other approaches.

needed. In consequence, any improvement from our method is not a direct result of the transformer being a stronger representation learning architecture. Our semantic compositional data generation process performs nearly on par with the semantic compositional RL baseline. Critically, as we discuss next, the DiT can then generate data for new tasks and iteratively improve its own performance.

**Iterated generative performance** Our iterative self-improvement algorithm increases all architectures' success rates. The monolithic architecture improves by 17%, standard DiT by 15%, semantic DiT by 16%, and semantic compositional DiT by 22%—the semantic compositional architecture achieves the largest absolute improvement. These marked improvements indicate that the nature of compositional data is useful for out-of-distribution generation. As our approach can self-improve, it quickly outperforms the static RL baseline without any additional real training data. Note that we can view the threshold $\tau$ as a soft upper bound on success rate, since we generate data that enables as little as $\tau$ success rate per task, and it is challenging to train policies that outperform this level of data quality. With $\tau$ reaching 0.7 at iteration four and our semantic compositional DiT achieving a success rate of 55%, the gap to this soft upper bound closes.

## 4.3 Iterative Compositional Data Generation

Next, we investigate each round of the iterative procedure for data generation. In every round of Algorithm 1, we evaluate five runs of TD3-BC for each unsolved task (sr $< \tau$) to average out the randomness from RL training and track the best success rate so far for each task. Figure 6 reports the average success rate in every iteration and the number of tasks that achieve success at least once (i.e. sr $> 0$).

**Success rate over time** Figure 6 (top left) reiterates the finding that all architectures consistently improve when artificial data is added. All architectures improve at a similar rate, and so the fact that only semantic architectures eventually outperform the RL version of our architecture is largely due to their significantly higher initial success rate. Our compositional semantic architecture only requires one round of self-improvement to exceed the performance of its RL counterpart. This interplay between initial generative performance and downstream RL performance highlights the importance of studying the two in tandem.

To assess whether these improvements reflect stable learning rather than an artifact of the monotonic best-so-far metric, we additionally report the raw per-iteration mean success rate in Figure 6 (top middle). For tasks that exceed the quality threshold and are removed from further generation rounds (Algorithm 1, Line 8), we carry forward the success rate achieved when the threshold is satisfied for subsequent iterations. Unlike the best-so-far curves, this metric reflects the actual performance achieved in each iteration and is not monotonic by design. We observe that the mean success rate increases across iterations for all architectures, with our compositional semantic architecture exhibiting the largest gains. This trend closely mirrors the

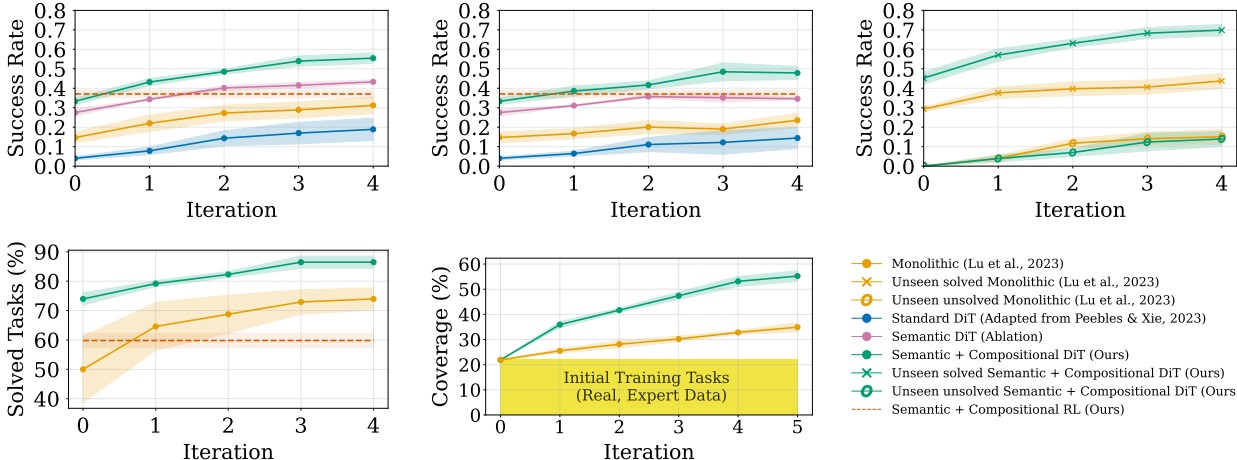

Figure 6: Performance of different diffusion architectures over iterations of our self-improvement procedure. (Top Left) Best success rate achieved so far, evaluated on held-out tasks. RL agents trained on synthetic data from the semantic compositional diffusion architecture consistently achieve higher success rates than agents trained on data from all other diffusion architectures and quickly surpass the semantic compositional RL baseline. (Top Middle) Raw per-iteration mean success rate, evaluated on held-out tasks. All architectures consistently improved success rate across iterations, indicating self-improvement is stable rather than accumulative. (Top Right) Best success rate achieved so far, evaluated on held-out tasks, separated by initial task difficulty. Tasks are partitioned into those that exhibit non-zero success at iteration 0 and those that were entirely unsolved. The semantic compositional architecture particularly improves performance on tasks on which it already obtains initial success in iteration 0. (Bottom Left) Number of tasks solved at least once across iterations. RL trained on synthetic data from the semantic compositional diffusion transformer outperforms both baselines and solves nearly all tasks at least once. (Bottom Middle) Task coverage over iterations. Coverage denotes the fraction of tasks (real + admitted synthetic datasets) included in the training pool relative to all possible task combinations. Synthetic datasets are added only after exceeding the quality threshold $\tau$ (Algorithm 1, Line 8). Starting from $\tau = 0.8$ (generated dataset achieved $> 80\%$ success rate), the semantic compositional model reduces the threshold to 0.7 by iteration 5, whereas the monolithic baseline reduces it to 0.6; the semantic compositional model attains higher coverage while maintaining higher synthetic training data quality. Shaded regions indicate standard error over 3 diffusion seeds.

improvements observed in the best-so-far success metric, indicating that the gains are not solely due to accumulating earlier successful tasks but reflect genuine improvements in the generator across iterations.

**Solved tasks over time** Figure 6 (bottom left) shows that the semantic compositional approach generates data that yields at least one successful trajectory more consistently than the monolithic approach. In addition, after four iterations of refinement, we solve almost every task at least once. This suggests that our approach could serve as a powerful starting point for fine-tuning new policies, since it can drastically reduce the exploration challenge of online RL. Interestingly, while RL using our architecture achieves a higher zero-shot success rate than our generative approach at iteration 0, this success rate is concentrated on a smaller fraction of tasks. This suggests that the diffusion model generalizes more broadly across tasks, but the initial data quality is insufficient to extract good policies in all parts of the state space.

To better understand whether tasks that are solved at least once correspond to only occasional successes or to consistently useful datasets, we further analyze the coverage of the training pool across iterations (Figure 6, bottom middle). Coverage measures the fraction of task combinations (real and admitted synthetic datasets) included in the training set relative to all possible combinations. Importantly, synthetic datasets are only admitted once they exceed the quality threshold $\tau$ (Algorithm 1, Line 8), ensuring that datasets generated from sporadic or low-quality successes are not incorporated into subsequent training rounds. As a result,

increasing coverage indicates that reliable datasets become available for a larger portion of the task space. We observe that the semantic compositional model increases coverage from roughly 20% to nearly 55%, while the monolithic baseline expands coverage more modestly to around 35%, resulting in an approximately 20% coverage gap after five iterations. This indicates that the compositional model produces reliable training datasets for a substantially larger portion of the task space. For the small number of tasks where synthetic datasets contain only rare successful trajectories, we further analyze whether such datasets can still provide useful signal for downstream reinforcement learning in Section 4.6.

Given that the semantic compositional model yields the most significant improvement in total success but also has a smaller improvement on tasks solved, one might conclude that our architecture is better at iteratively deriving information from successful tasks by refining marginals. To verify this, we analyze whether iterative improvements appear on tasks that see some success or tasks that are not yet solved. In Figure 6 (top right), we show that the monolithic architecture improves roughly at the same rate on both the already successful and unsuccessful tasks. While our semantic compositional model also improves in both regimes, it obtains a much larger jump in performance on tasks that see some initial success at iteration 0. In part, this stems from the fact that there are few tasks left on which no success is achieved initially. Yet, it also provides evidence that our semantic compositional model improves encoder-decoder pairs point-wise using self-generated data.

## 4.4 Analyzing Compositional Structure

This section studies the compositional structure learned by our architecture. As discussed in Section 2.2, we use the Elucidated Diffusion approach (Karras et al., 2022), which parameterizes the diffusion process using continuous noise levels $\sigma$ rather than discrete timesteps, with default noise range $\sigma \in [\sigma_{\min}, \sigma_{\max}]$. For our analysis, we evaluate the model's behavior at a noise level $\sigma_{\mathrm{midpoint}}$ corresponding to the midpoint of the generation schedule, computed per the sampling schedule formula (Karras et al., 2022):

$$
\sigma_{i<N} = \left( \sigma_{\max}^{1/\rho} + \frac{i}{N-1} \left( \sigma_{\min}^{1/\rho} - \sigma_{\max}^{1/\rho} \right) \right)^{\rho} \;, \quad \sigma_N = 0 \;,
$$

where $i$ denotes the step. $\sigma_{\mathrm{midpoint}}$ represents a moderate noise level the model encounters during generation. Throughout this section, we use the DiT trained at iteration 0, using only real data.

**Intervention influence**  To analyze the compositional dependencies that our model learns, we compute an influence matrix that measures how inputs to each encoder module affect the outputs of each decoder module. For a given task, we generate random Gaussian noise inputs and compute the outputs at $\sigma_{\mathrm{midpoint}}$. We then systematically intervene on each encoder module by zeroing out its output patches and measure the resulting change in each decoder module's output. The influence of encoder module $i$ on decoder module $j$ is quantified as the $L_2$ norm of the normalized difference between the intervened and nominal decoder outputs. Averaging these normalized differences across many noise samples yields an influence matrix whose entries quantify the causal effect of one module on another while remaining comparable across decoders of different dimensionality. This allows us to measure, for example, which predictions are most affected if the object information is missing. Figure 7 presents the average intervention influence matrix over all training tasks.

As expected, the largest deviations happen on the diagonal, as masking a certain element at the encoder makes it difficult to accurately generate that element itself. For example, if the object embedding is masked, the transformer relies exclusively on task conditioning to generate object information. Variations across state components further expose a particular dependency structure among elements. For instance, task input influences object prediction more than it does obstacle prediction. Yet all components depend on each other to some degree. This is not particularly surprising, since the state representation contains relative information (e.g., relative poses). More interestingly, many decoder outputs rely heavily on the robot arm encoding. This highlights the crucial importance of the robot arm in our model for generating data. The largest influence outside the diagonal is for the robot arm input and reward prediction. In general, the reward predictions greatly depend on the state components but not so much on the action, which is correctly inferred by the model since the reward in CompoSuite is only a function of the state. The terminal signal

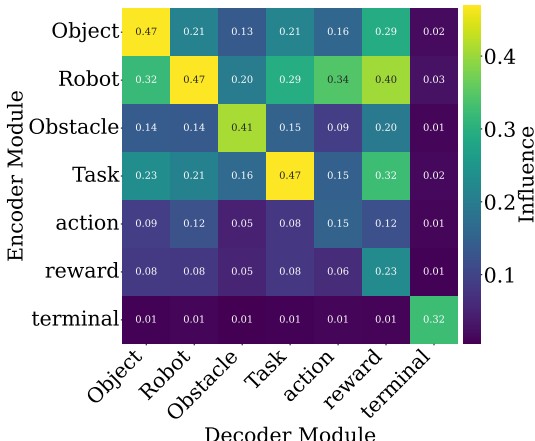

Figure 7: Average intervention influence over all training tasks. The heatmap shows how masking each encoder module (rows) changes the predictions produced by each decoder module (columns). Brighter values indicate a larger influence. The plot shows strong diagonal effects, meaning each factor depends most on its own encoder, but it also reveals notable cross-factor interactions. In particular, the robot encoder has a strong effect on several other decoders, especially the reward decoder.

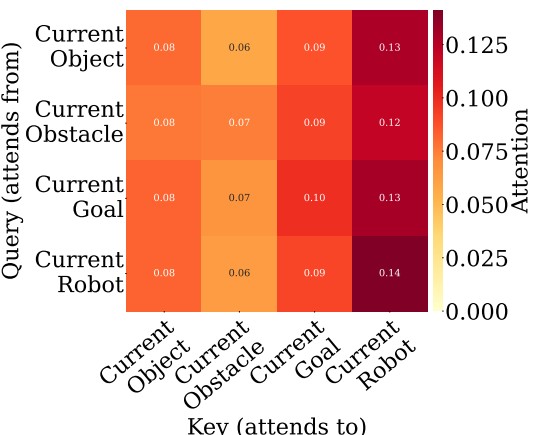

Figure 8: Average attention weights over all training tasks for a single block diffusion transformer. The heatmap shows how much each state component encoder (rows) attends to every other component (columns) when generating denoised representations. Darker colors indicate stronger attention. The plot shows a clear ordering in which the robot component receives the strongest attention from all queries, followed by the goal, the object, and finally the obstacle.

depends exclusively on its own input, since the dataset released by Hussing et al. (2024) contains expert trajectories for almost all tasks, making failure terminations rare in the training data.

**Attention masks** Now, we shift our focus to the state encoder structure, the central piece of our architecture design. We study the DiT attention structure by capturing the full 11×11 self-attention matrices. To inspect which encoder outputs map to which decoders, we trained a single-layer transformer. We draw 100 Gaussian inputs, run the model at $\sigma_{\mathrm{midpoint}}$, and compute per-head attention weights from every Multi-Head Self-Attention block. We do this for the 14 training tasks and average across samples and attention heads. We are mostly interested in the state decomposition, which is the main distinguishing feature of our architecture. Figure 8 shows the entries of the attention matrix corresponding to state elements.

The state attention mask reveals that there exists a non-trivial mapping between the state encoder and decoder pairs. First, every decoder pays some attention to its corresponding encoder (the diagonal). Then, we observe an ordering of importance across state elements. Every encoder pays greatest attention to the robot, then the objective, then the object, then the obstacle. The ordering we find is contrary to the hard-coded architecture of Mendez et al. (2022a), where the robot modules are stacked onto the remaining modules last, implying that other encoders cannot access robot information. This difference may be stem from fundamentally different computations required to learn an RL policy compared to generating RL data.

To verify that this dependency structure is not limited to the single-layer analysis above, we extend the attention analysis to the full diffusion transformer and across all iterations of self-improvement. Figure 9 shows the attention patterns for the state components across the eight transformer layers and five training iterations. Similar to the single-block analysis, we observe consistent diagonal structure, indicating that each decoder continues to attend to its corresponding encoder. More importantly, the relative importance ordering between components remains stable throughout training: across layers and iterations, the robot component consistently receives the strongest attention from all queries, followed by the goal, while the object and obstacle receive comparatively weaker attention.

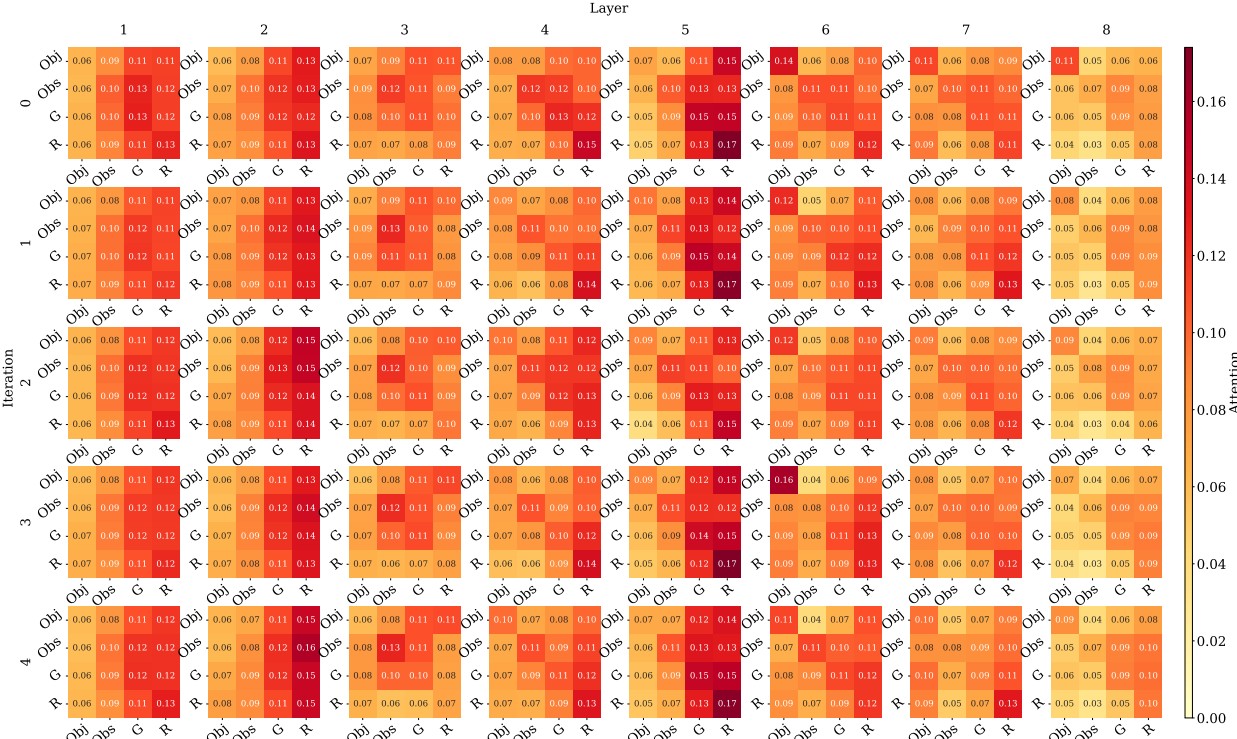

Obj: Current Object,  Obs: Current Obstacle,  G: Current Goal,  R: Current Robot,  x-axis: Key (attends to),  y-axis: Query (attends from)

Figure 9: Average attention weights over all training tasks across transformer layers and self-improvement iterations. Each heatmap shows how much each state component encoder (rows) attends to every other component (columns) when generating denoised representations. Columns correspond to transformer layers and rows correspond to diffusion training iterations. Darker colors indicate stronger attention. Across iterations, the learned dependency structure remains stable and becomes slightly more pronounced, with the robot component receiving the strongest attention, followed by the goal, while the object and obstacle receive comparatively weaker attention.

While the overall ordering remains stable, the attention patterns become slightly more pronounced in the middle layers of the transformer (layers 4–6), suggesting that compositional interactions between components might be primarily captured in intermediate representations. Importantly, the dependency structure learned at iteration 0 is preserved throughout subsequent self-improvement iterations, indicating that iterative training does not disrupt the learned compositional relationships between task components. These results provide further evidence that the transformer consistently learns and maintains the underlying dependency structure across both depth and training iterations.

## 4.5 Environment Interaction Efficiency of Iterative Compositional Data Generation

While our approach reduces the need to collect expert demonstrations for unseen tasks, the iterative data generation procedure still requires environment interaction to evaluate policies trained on generated datasets. In this section, we quantify this interaction cost and analyze its efficiency relative to a strong offline-to-online reinforcement learning baseline. We additionally report computational requirements and runtime of the iterative compositional data generation in Section A.4.

Algorithm 1 evaluates candidate datasets by training policies on generated transitions and performing short rollout episodes in the environment. Each evaluation consists of 10 trajectory rollouts of length 500, corresponding to approximately 5,000 environment transitions per task per iteration. Across four iterations,

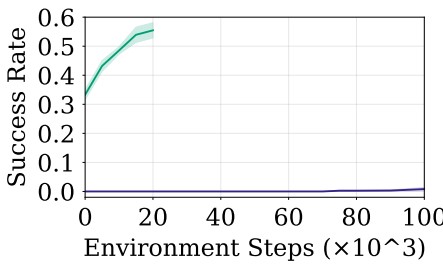 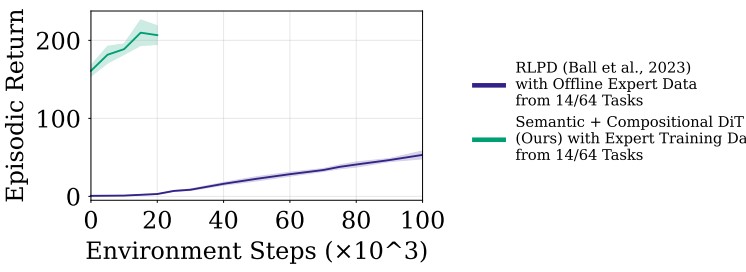

Figure 10: Environment interaction efficiency comparison on 32 held-out tasks. (Left) Best success rate as a function of environment interactions. (Right) Mean episodic return as a function of environment interactions. RLPD (Ball et al., 2023) performs online reinforcement learning on the target task using up to 100,000 environment interactions while leveraging expert data from 14 training tasks. Our iterative compositional data generation approach achieves substantially higher performance using only 20,000 environment interactions. Shaded regions indicate standard error over 3 diffusion seeds for Semantic + Compositional DiT, and 5 RL seeds for the RLPD baseline.

this amounts to roughly 20,000 environment interactions per task. Importantly, these interactions are used only to evaluate policy performance and determine whether a generated dataset should be incorporated into training; they are not used to update the policy or the generative model.

Meanwhile, the iterative compositional data generation pipeline produces very large synthetic datasets. For each unseen task, the generator produces approximately 2,000 rollouts of length 500, corresponding to roughly 1 million synthetic transitions. These trajectories are produced entirely by the diffusion model and can be generated in parallel without additional environment interaction. Collecting datasets of this scale through human teleoperation would require substantial manual effort and repeated environment resets, making large-scale expert data collection impractical.

To assess interaction efficiency, we compare against RLPD (Ball et al., 2023), an off-policy reinforcement learning algorithm that combines offline datasets with online experience. RLPD initializes learning with a replay buffer containing offline trajectories and continues to collect new transitions online, training the policy using minibatches sampled from both sources. Both approaches start from the same initial dataset consisting of expert demonstrations from 14 training tasks. For each held-out target task, RLPD interacts with the environment and updates the policy using 100,000 environment steps while leveraging the offline datasets from the 14 expert training tasks.

Figure 10 shows the resulting mean episodic return and best success rate across 32 held-out test tasks as a function of environment interaction. Despite directly training on the target task with 100,000 environment interactions, RLPD achieves near-zero success rate with an episodic return of roughly 55. In contrast, our iterative compositional data generation approach reaches a success rate above 55% and an episodic return exceeding 200 using only 20,000 environment interactions after four improvement iterations.

These results demonstrate the efficiency of the proposed approach. A small number of evaluation interactions suffices to validate large synthetic datasets generated by the compositional diffusion model. While our method requires access to an environment or simulator for evaluation, these interactions are autonomous, do not update the policy, and can be executed in parallel. As a result, their cost is substantially lower than collecting additional expert demonstrations, which would require expert policies or human teleoperation for each new task. In addition, one can imagine to find a different proxy for reward quality that determines whether datasets are added inside our procedure that does not require environment interaction.

## 4.6 Utility of Rare Successful Synthetic Trajectories

While Section 4.3 demonstrates that datasets admitted into the training pool correspond to reliable task solutions, a small subset of tasks may still yield synthetic datasets with relatively low success rates. An

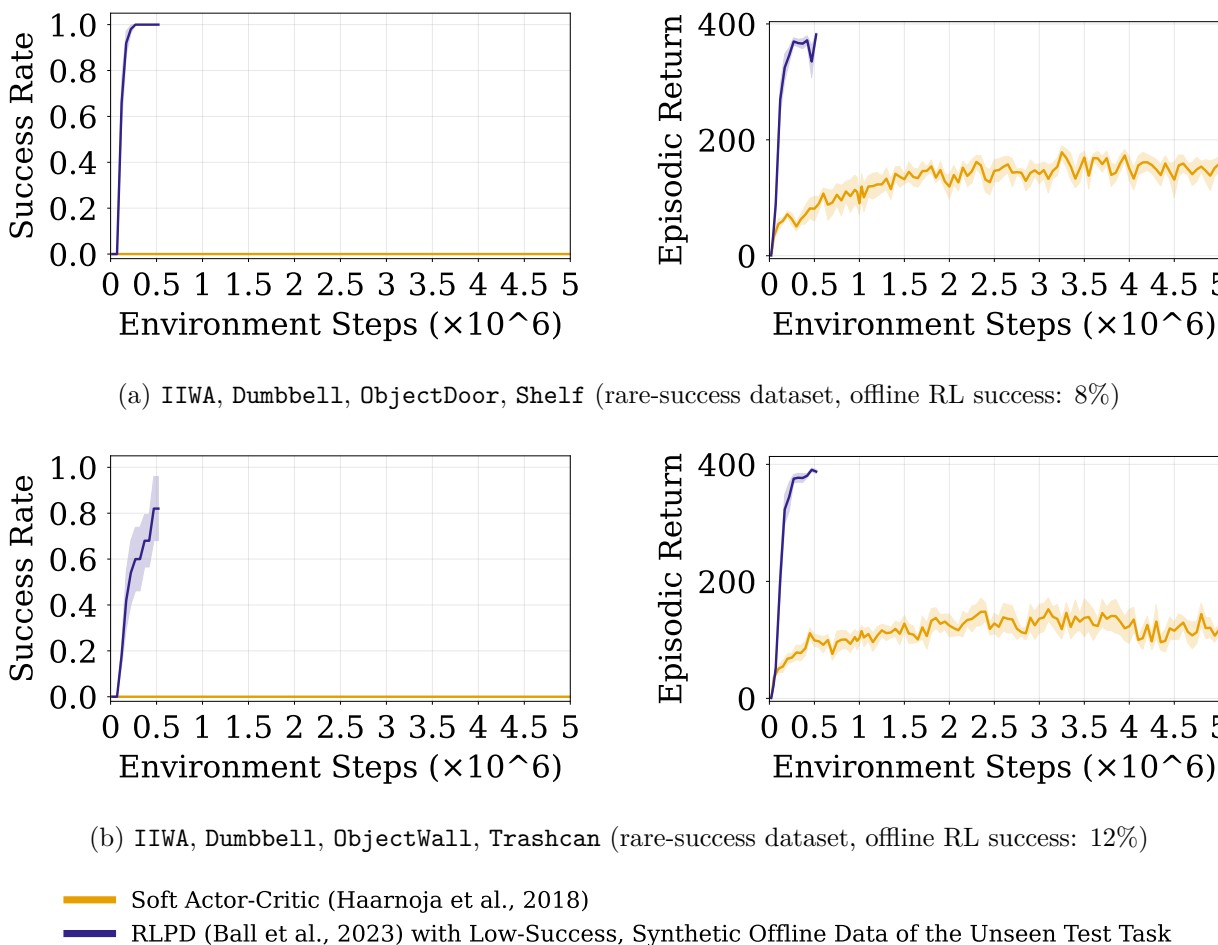

(a) `IIWA`, `Dumbbell`, `ObjectDoor`, `Shelf` (rare-success dataset, offline RL success: 8%)

(b) `IIWA`, `Dumbbell`, `ObjectWall`, `Trashcan` (rare-success dataset, offline RL success: 12%)

— Soft Actor-Critic (Haarnoja et al., 2018)
— RLPD (Ball et al., 2023) with Low-Success, Synthetic Offline Data of the Unseen Test Task

Figure 11: Utility of rare successful synthetic trajectories for bootstrapping online reinforcement learning. (Left) Best success rate and (Right) episodic return as a function of environment interactions. We select the two synthetic datasets with the lowest non-zero success rates produced by the iterative compositional generation process (seed 0), where dataset success rate is measured by training an offline reinforcement learning policy using TD3-BC on the dataset. (a) `IIWA`, `Dumbbell`, `ObjectDoor`, `Shelf` with dataset success rate 8%. (b) `IIWA`, `Dumbbell`, `ObjectWall`, `Trashcan` with dataset success rate 12%. RLPD is initialized with the corresponding synthetic dataset of the target task and performs online reinforcement learning, while SAC is trained from scratch without offline data. Despite the extremely low dataset success rates, RLPD rapidly solves both tasks, reaching success rates between 80% and 100% within fewer than 500,000 environment interactions, whereas SAC fails to achieve any successful episodes even after more than 5 million interactions. Shaded regions indicate standard error across five seeds.

important question is therefore whether such datasets, which contain only occasional successful trajectories, can still provide useful signal for downstream reinforcement learning.

To study this, we analyze the utility of synthetic datasets with extremely low success rates. Specifically, we select two datasets with the lowest non-zero success rates produced by the iterative compositional data generation process (seed 0), with dataset success rates of 8% and 12% when evaluated using TD3-BC. These datasets correspond to tasks where successful trajectories occur only rarely. We initialize the replay buffer of RLPD with a single synthetic dataset corresponding to the target unseen test task and perform online reinforcement learning. For comparison, we also train a standard Soft Actor-Critic (SAC) (Haarnoja et al., 2018) agent from scratch without access to this dataset.

Figure 11 shows the resulting learning curves. Despite the extremely low dataset success rates (8% and 12%), RLPD rapidly solves both tasks, reaching success rates between 80% and 100% within fewer than 500,000 environment interactions. In contrast, SAC trained from scratch fails to achieve any successful episodes even after more than 5 million environment interactions. The plotted curves also account for the environment interactions required during dataset evaluation in the iterative generation process, which amount to approximately 20,000 interactions per task. As shown in the figure, this additional evaluation cost is negligible relative to the overall training horizon.

These results demonstrate that even datasets containing only occasional successful trajectories can provide sufficient signal to substantially accelerate online reinforcement learning. This observation highlights the practical importance of broad task coverage: although individual datasets may exhibit low success rates, their availability across many tasks can still provide valuable priors that enable efficient online adaptation. Consequently, even rare successful trajectories can significantly improve the efficiency of downstream learning.

## 4.7 Scaling to Larger Compositional Task Spaces

### 4.7.1 Justifying the Low-Data Regime to Study Compositionality

When sufficient expert data is available, standard feed-forward policies trained with behavioral cloning on the CompoSuite datasets achieve non-trivial zero-shot generalization (Hussing et al., 2024). However, this assumes access to expert trajectories for hundreds of tasks, which is unrealistic in many robotics applications. As data becomes sparser, exploiting the compositional structure of the tasks becomes more relevant. Here, we verify that the data regime of 14 training tasks from Sections 4.2–4.6 is appropriate for studying compositionality. Using all 10 task-lists from the experimental setup suggested by Hussing et al. (2024), we construct subsets of training tasks using the first $N$ tasks from each list, where $N \in \{56, 98, 140, 182, 224\}$, keeping the set of 32 test tasks fixed across values of $N$. We then train the SynthER-based architecture introduced in section 4.1 on each subset of training tasks. We generate one million transitions for each test task and train a per-task TD3-BC agent on the generated data. We measure the difference in accumulated return over a set of evaluation trajectories relative to a TD3-BC agent trained on real data. We expect that when the amount of available training data becomes small, the TD3-BC performance should decrease

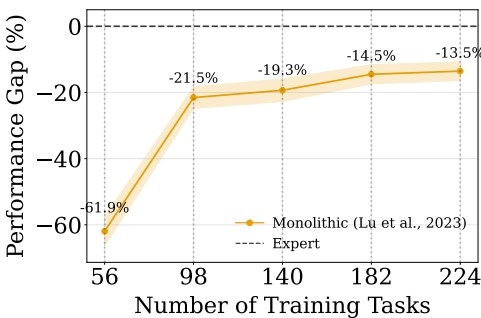

Figure 12: Return difference between RL policies trained on data generated by the monolithic architecture and policies trained on ground-truth data over varying number of training tasks. As the number of training tasks approaches 56 ($\sim 20\%$), there is steep increase in the performance gap indicating the sub-optimality of generated data from the diffusion model.

as the generated data quality on out-of-distribution tasks decreases. We report the results in Figure 12.

The results show that when more than 182 tasks are available for training the diffusion models, the mean gap to the ground-truth policy performance is less than 15%. While even the diffusion model trained on 98 tasks achieves high zero-shot generalization, we see a downward trend below this point. As expected, when we move to 56 tasks (roughly 20% of the tasks) the performance gap increases drastically, and the model is unable to zero-shot generalize meaningfully. This is a similar data regime to the 14/64 tasks we used to show the ability of our compositional DiT to learn the underlying graph compositional structure.

### 4.7.2 Results at Larger Scale

To further validate that this regime exhibits behavior consistent with our IIWA-only setting, we repeat the iterative data generation procedure in the 56/256-task regime and directly compare the monolithic SynthER-style diffusion model with our semantic compositional diffusion transformer architecture. Figure 13 reports the evolution of zero-shot success rates and solved-task coverage over iterations of Algorithm 1.

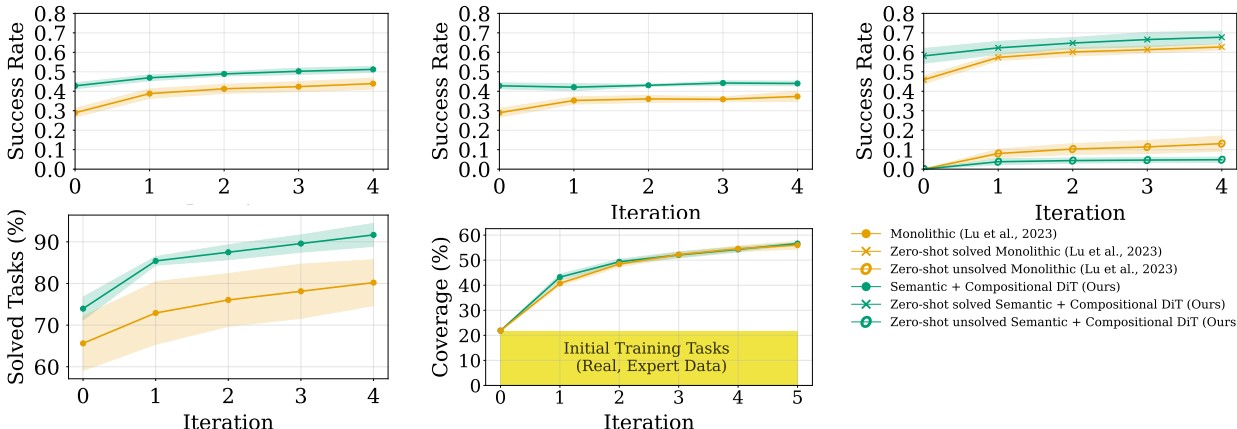

Figure 13: Performance of the monolithic SynthER-based diffusion architecture and the semantic compositional diffusion transformer over iterations of the self-improvement procedure in the 56/256-task training regime. (Top Left) Best success rate achieved so far, evaluated on held-out tasks. RL agents trained on synthetic data from the semantic compositional diffusion architecture consistently achieve higher success rates than agents trained on data from the monolithic SynthER-style diffusion architecture. (Top Middle) Raw per-iteration mean success rate, evaluated on held-out tasks. Both architectures slightly improved success rate across iterations, indicating self-improvement is stable rather than accumulative. (Top Right) Best success rate achieved so far, evaluated on held-out tasks, separated by initial task difficulty. Tasks are partitioned into those that exhibit non-zero success at iteration 0 and those that were entirely unsolved. The semantic compositional architecture particularly improves performance on tasks for which some initial success is already obtained. (Bottom Left) Number of tasks solved at least once across iterations. The semantic compositional diffusion transformer enables policies to solve a substantially larger fraction of tasks at least once compared to the monolithic baseline. (Bottom Middle) Task coverage over iterations. Coverage denotes the fraction of tasks (real + admitted synthetic datasets) included in the training pool relative to all possible task combinations. Synthetic datasets are added only after exceeding the quality threshold $\tau$ (Algorithm 1, Line 8). Starting from $\tau = 0.8$ (generated datasets achieving $> 80\%$ success), the semantic compositional model maintains the threshold throughout, whereas the monolithic baseline reduces it to 0.7. Although coverage is comparable, the semantic compositional model maintains higher synthetic training data quality, which contributes to its superior performance in the other panels. Shaded regions indicate standard error over 3 diffusion seeds.

We observe similar trends to those in Figure 6: both architectures benefit from iterative self-improvement, while policies trained on data generated by the semantic compositional model consistently achieve higher zero-shot success rates and solve a larger fraction of tasks at least once across iterations. Notably, the performance gap between the monolithic and semantic compositional model is slightly reduced in the 56/256 setting. This is expected, as increasing the number of compositional axes while preserving a comparable training-task ratio exposes the monolithic model to a broader set of task combinations, partially alleviating the severity of the generalization challenge. Importantly, the compositional model continues to exhibit a clear advantage, indicating that the benefits of exploiting task structure persist beyond the IIWA-only setting.

Separating tasks by initial difficulty reveals a similar asymmetry to that observed in the main experiments. While the semantic compositional model improves in both regimes, it exhibits substantially larger gains on tasks that achieve non-zero success at iteration 0, with more gradual improvements on initially unsolved tasks. This behavior is consistent with the interpretation that the model refines factor-specific encoder–decoder representations using self-generated data.

The task coverage dynamics further support the comparability of the two regimes. In the 56/256 setting, both architectures attain similar coverage; however, the semantic compositional model maintains the initial quality threshold ($\tau = 0.8$), while the monolithic baseline reduces it to 0.7. This difference in retained

data quality is reflected in the consistently stronger performance of the semantic compositional model across the other evaluation metrics. At the same time, when considering the 14/64 IIWA-only regime from the main paper, coverage expansion diverges substantially across methods. Although both regimes use a similar training-task ratio (approximately 20%), the 14/64 setting involves only 14 observed task combinations in absolute terms, resulting in more limited raw task diversity. Under this increased combinatorial sparsity, exploiting task structure becomes more critical, which explains the amplified advantage of the semantic compositional model in the main experiments. Thus, while the 56/256 results confirm consistent qualitative behavior, the 14/64 regime constitutes a more demanding compositional generalization scenario despite using a comparable proportion of training tasks.

Overall, these results support the conclusion that the 14/64 IIWA-only setting constitutes a particularly challenging low-data regime, and that the qualitative self-improvement dynamics of the semantic compositional diffusion model are stable as the number of compositional task axes increases.

## 5   Related Work

**Compositional Generalization in Robotics and RL**   In robotics, compositional generalization has been pursued through a variety of mechanisms. Some approaches introduce modularity or architectural biases aimed at composing semantic units such as instructions or high-level skills (Xu et al., 2018; Devin et al., 2019; Kuo et al., 2020; Wang et al., 2023). Other work targets the control layer directly, designing modular, factorized, or entity-centric policy architectures that encourage reuse of behavioral components across tasks (Devin et al., 2017; Mendez et al., 2022b; Zhou et al., 2025). Some approaches seek to automatically identify and decompose policies into functional modules (Yang et al., 2020; Mittal et al., 2020; Goyal et al., 2021). A complementary direction exploits scene-centric formulations that use structured object-relational representations to compose low-level visuomotor skills in novel physical configurations (Qi et al., 2025).

These approaches demonstrate the importance of leveraging task structure, but they typically assume a *hand-designed* decomposition of robots, objects, and goals. In contrast, our method learns this structure *directly from data* by interpreting a diffusion transformer as a GNN and equipping it with factor-specific tokenizers. Whereas prior work uses compositionality to structure *policies*, we instead use it to structure a *generative model of transitions*, enabling zero-shot synthesis of data for unseen task compositions.

**Generated Data in Robotics and RL**   Synthetic data has become a key idea for scaling robotic learning. One line of work expands imitation datasets through trajectory-level augmentations, where expert demonstrations are perturbed, resampled, or regenerated to increase coverage (Mandlekar et al., 2023; Wang et al., 2024; Ameperosa et al., 2025; Jiang et al., 2025). Although these methods enrich demonstration sets, they remain constrained to variations of the same underlying tasks without expanding into *new compositions*.

Complementary efforts in reinforcement learning explore *generative replay*, where learned generative models synthesize transitions to supplement or replace entries in an agent's replay buffer (Huang et al., 2017; Ludjen, 2021; Imre, 2021; Lu et al., 2023; Voelcker et al., 2025). As generative modeling techniques have advanced—from variational autoencoders (Kingma & Welling, 2014) and generative adversarial networks (Goodfellow et al., 2014) to, more recently, diffusion models (Karras et al., 2022)—the fidelity of replayed experience and the sample efficiency of these approaches have improved accordingly. However, these methods still generate data only for the *same tasks* observed during training. They do not attempt to produce transitions for *unseen combinations of factors* that fall outside the original task distribution.

Another orthogonal direction emphasizes visual augmentation, including render-driven and vision-only pipelines that procedurally generate synthetic video datasets (Bonetto et al., 2023; Singh et al., 2024; Yu et al., 2024; Han et al., 2025; Yu et al., 2025), as well as generative and diffusion-based methods that augment images while holding actions constant (Chen et al., 2023; Yu et al., 2023). While effective for increasing visual diversity, these approaches do not provide transition-level data reflecting novel task semantics.

Our work is complementary to all of these efforts but differs fundamentally: we generate full state-action-next-state transitions for *unseen* tasks. Moreover, our iterative procedure evaluates the usefulness of generated data via offline RL, creating a closed-loop mechanism for self-improving compositional data generation.

**Compositional Data Generation in Robotics** Recent work has shown that exposing robots to compositional factors of variation can improve generalization and reduce the amount of manually collected data (Gao et al., 2024). In parallel, generative approaches have begun to exploit compositional structure to synthesize additional training experience (Zhou et al., 2024; Barcellona et al., 2025). These works demonstrate the value of decomposing environments into reusable components, but they study compositionality in different settings from ours.

Gao et al. (Gao et al., 2024) analyze compositional generalization in *data collection strategies*, showing that imitation policies can generalize to unseen combinations of environmental factors when demonstrations are collected strategically. However, their work focuses on how to collect demonstrations for a fixed task and does not generate synthetic data or model transition dynamics.

Other approaches employ compositional generative models in pixel space. For example, Dream to Manipulate (Barcellona et al., 2025) augments demonstrations by applying equivariant transformations to existing trajectories, increasing data diversity while keeping the task and goal fixed. RoboDreamer (Zhou et al., 2024) learns a compositional video world model and evaluates it through imagination-based planning, where generated videos are mapped to actions via inverse dynamics. In both cases, compositionality is used to generate diverse observations or demonstrations within existing tasks.

In contrast, our work studies *inter-task compositional generalization*. Rather than augmenting demonstrations for a fixed task, we model the transition distribution for unseen compositions of task components and use the generated transition data to train control policies. To our knowledge, the closest related approach is SynthER, which also studies generative modeling of transition data for offline RL. However, SynthER focuses on improving policy learning within existing tasks, whereas our method generates data for entirely new task compositions and iteratively improves the generative model using validated synthetic datasets.

## 6 Limitations and Future Work

While reducing data collection is a motivating goal for this line of work, we do not evaluate our method in real-world robotic settings in this paper. Assessing its performance in such settings remains an important direction for future work, especially in large-scale compositional settings that require evaluation protocols capable of supporting the scale considered in our setup. Our experiments involve up to hundreds of task combinations, for example 256 tasks, whereas most real-robot studies evaluate on substantially smaller task sets. Extending this evaluation to physical robots would therefore likely require multiple platforms, diverse configurations, and repeated evaluations across tasks and iterations of the self-improvement loop.

At present, our method decides whether a generated task dataset is added to the training pool by running an online evaluation loop. We deploy an RL agent on the newly generated task and include the corresponding data only if its success rate exceeds a fixed threshold. This makes our procedure dependent on interaction with the environment, which can be costly in many real-world settings. Note that our approach is not tied to this particular choice and any suitable scoring function to assess the utility of generated data could be used instead. Replacing this online evaluation with interaction-free or partially offline proxies that can reliably predict the utility of newly generated data is a key direction to further improve the efficiency of the approach.

Finally, our current approach operates on state-based representations. Extending this framework to visual observations is another important avenue to broaden its applicability, for example by incorporating structured perception modules (e.g., segmentation or object-centric representations) that provide a similar factorization of visual inputs and allow compositional reasoning to be applied in image-based settings.

## 7 Conclusion

In this work, we introduce an iterative compositional data generation framework that uses a semantic compositional diffusion transformer and a self-improvement loop to synthesize and curate manipulation data. This data is of sufficient quality to train policies that solve novel combinations of compositional tasks. More broadly, our work addresses a central challenge in robotics: scaling learning to a large number of diverse and compositional tasks. Collecting data for every task variation is prohibitively expensive, and our results high-

light that learning compositional structure directly from limited data enables effective reuse of interactions to self-generate training data for novel task combinations. Given the current state-based formulation of our method, this approach is particularly amenable to settings where factorized representations are available, enabling scalable policy learning. In particular, it aligns well with regimes that utilize privileged or teacher policies, which can access full state information during training and later be distilled into policies operating under partial observations for sim-to-real transfer.

### Acknowledgments

DSB acknowledges support from the Center for Curiosity. MH and EE were partially supported by the Army Research Office under MURI award W911NF201- 0080 and DARPA award HR00112420305. Any opinions, findings, conclusions, or recommendations expressed in this material are those of the authors and do not necessarily reflect the view of DARPA, the US Army, or the US government.

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

# A Additional Experimental Details

This section provides details of the experimental setting used to obtain all results in the paper.

## A.1 Compositional Data Generation

### A.1.1 Diffusion Model Training

We train diffusion models on transition tuples $(s, a, r, s', d)$ with total dimension 164: a 77-dimensional state, an 8-dimensional action, a scalar reward, a 77-dimensional next state, and a binary terminal indicator. All continuous dimensions are standardized (zero mean, unit variance) using statistics computed from the pooled dataset of all training tasks. The terminal indicator remains unnormalized and is discretized with a threshold of 0.5. All model variants use the same Elucidated Diffusion schedule; differences arise only from the denoiser architecture and a small number of optimization hyperparameters.

During training, noise levels are sampled from a log-normal distribution with mean $P_{\mathrm{mean}} = -1.2$ and standard deviation $P_{\mathrm{std}} = 1.2$. The loss weighting function uses $\sigma_{\mathrm{data}} = 1.0$, and the noise schedule spans $\sigma \in [0.002, 80]$ with curvature parameter $\rho = 7$. Tasks are encoded as 16-dimensional binary indicator vectors, as illustrated in Figure 3, and these task indicators condition the denoiser during both training and generation. The complete set of architectural and optimization hyperparameters for the **Monolithic** baseline (Lu et al., 2023) and our **Semantic + Compositional DiT** (S+C DiT) model is listed in Table 1.

### A.1.2 Data Generation

After each diffusion model is trained, we generate synthetic transition datasets for individual tasks using the EMA (Exponential Moving Average) version of the model. Each task is specified as a 4-tuple $(\mathrm{Robot}, \mathrm{Object}, \mathrm{Obstacle}, \mathrm{Objective})$ and encoded using the same 16-dimensional task indicator from Figure 3. The Monolithic and S+C DiT pipelines use identical sampling configurations, except for the generator batch size, which is reduced for S+C DiT to satisfy GPU memory constraints. The hyperparameters used for synthetic data generation are summarized in Table 2.

### A.1.3 Policy Training

We train TD3-BC policies (Fujimoto & Gu, 2021) on the synthetic transition datasets generated for each task. The same TD3-BC configuration is used across all experiments, including both the monolithic and S+C DiT pipelines and all iterations. The only exception is the compositional RL baselines, which use different compositional policy architectures that are described in Appendix A.2. All policies are trained offline on synthetic data and are then evaluated online in the corresponding CompoSuite environment.

For all test tasks, we train policies using five random seeds and report the mean success rate. States are normalized using the mean and standard deviation computed from each task's synthetic training dataset. The complete set of TD3-BC hyperparameters used in all experiments is provided in Table 3.

### A.1.4 Iterative Bootstrapping

The iterative bootstrapping procedure follows Algorithm 1. We initialize the success threshold at $\tau_0 = 0.8$. If no new tasks satisfy this threshold for one iteration ($C = 1$), the threshold is automatically reduced by $\Delta_\tau = 0.1$, with a lower bound of $\tau_{\mathrm{min}} = 0.5$. The IIWA-only task list used for this experiment is described in Appendix A.5. The complete set of hyperparameters for this procedure is reported in Table 4.

## A.2 Compositional RL Baselines

We compare against two compositional RL baselines that train multitask TD3-BC policies on expert demonstrations from the same 14 training tasks used for diffusion training. Their train/test split matches the diffusion setup to allow a fair zero-shot generalization comparison. The list of training and test tasks is described in Appendix A.5. All baselines are evaluated on the corresponding 32 held-out test tasks using

Table 1: Diffusion model hyperparameters.

| COMPONENT | MONOLITHIC (Lu et al., 2023) | S+C DiT (Ours) |
|---|---|---|
| Architecture | 6-layer MLP, width 2048 | 8-layer DiT, hidden size 416, 8 heads (patch size 15 in standard DiT ablation) |
| Network capacity | 25.88M parameters | 26.22M parameters |
| Batch size | 1024 | 1024 |
| Learning rate | $3 \times 10^{-4}$ | $1 \times 10^{-4}$ |
| Weight decay | 0.0 | 0.01 |
| Optimizer | AdamW | AdamW |
| LR scheduler | Cosine | Cosine |
| Training steps | 100,000 | 100,000 |

Table 2: Synthetic data generation hyperparameters.

| COMPONENT | MONOLITHIC (Lu et al., 2023) | S+C DiT (Ours) |
|---|---|---|
| Generated samples per task | 1,000,000 | 1,000,000 |
| Sampling steps | 128 | 128 |
| Noise perturbation strength | 80 | 80 |
| Minimum noise level for perturbation | 0.05 | 0.05 |
| Maximum noise level for perturbation | 50 | 50 |
| Relative perturbation noise scale | 1.003 | 1.003 |
| Generator batch size | 100,000 | 25,000 |
| Batches per task | 10 | 40 |

15 random seeds. Both baselines follow the TD3-BC configuration in Table 3, but differ in three ways: (1) they train on a multitask dataset that combines demonstrations from all 14 training tasks, (2) they employ compositional policy architectures, and (3) their batch sizes are scaled with the number of training tasks (i.e., a multiple of the number of tasks), following the strategy used in Mendez et al. (2022a).

The **Hardcoded Compositional RL** (HC RL) baseline uses the modular architecture of Mendez et al. (2022a) with component-specific networks for each task element. The hardcoded architecture follows a hierarchical graph structure with the ordering Obstacle → Object → Subtask → Robot. Hidden layer sizes are (32) for Obstacle, (32, 32) for Object, (64, 64, 64) for Subtask, and (64, 64, 64) for Robot.

The **Semantic Compositional RL** (S+C RL) baseline uses a transformer with semantic tokens corresponding to the object, obstacle, goal, and robot. Compositional encoders produce token embeddings, and task conditioning is applied using Adaptive Layer Normalization (AdaLN). The transformer uses hidden size 72, depth 1, 4 attention heads, MLP ratio 1.20, and no dropout. For this baseline, the batch size is further reduced by one half compared to Mendez et al. (2022a) to satisfy GPU memory constraints.

The full set of hyperparameters is listed in Table 5.

### A.3 Online RL Experiments

#### A.3.1 RLPD with Real Expert Data

To quantify environment-interaction efficiency, we compare against RLPD (Ball et al., 2023), an offline-to-online RL algorithm that learns from a replay buffer containing both offline and online transitions. In this setting, the offline buffer is constructed from expert demonstrations on the same 14 training tasks used by our method. RLPD is then trained online on each held-out target task for up to 100,000 environment interactions, and we report results across all 32 held-out tasks over 5 seeds.

Table 3: TD3-BC offline RL training hyperparameters.

| COMPONENT | VALUE |
|---|---|
| Algorithm | TD3-BC |
| Actor network | MLP with 2 hidden layers of width 256 |
| Critic networks | MLP with 2 hidden layers of width 256 |
| Learning rate (actor, critics) | $3 \times 10^{-4}$ |
| Optimizer | Adam |
| Batch size | 1024 |
| Training steps | 50,000 |
| Discount factor ($\gamma$) | 0.99 |
| Target network update ($\tau$) | 0.005 |
| Regularization coefficient ($\alpha$) | 2.5 |
| Policy noise | 0.2 |
| Noise clip | 0.5 |
| Policy update frequency | 2 |
| Evaluation frequency | 5,000 steps |
| Evaluation episodes | 10 |
| State normalization | Yes |
| Reward normalization | No |
| Training seeds | 0–4 |

Table 4: Compositional iterative bootstrapping hyperparameters.

| COMPONENT | VALUE |
|---|---|
| Initial success threshold ($\tau_0$) | 0.8 |
| Minimum threshold ($\tau_{\min}$) | 0.5 |
| Threshold reduction amount ($\Delta_\tau$) | 0.1 |
| Patience ($C$) | 1 iteration |
| Training tasks ($|\mathcal{T}\text{train}|$) | 56 of 256 tasks, or 14 of 64 IIWA-only tasks |
| Diffusion seeds | 0–2 |

RLPD in this setting uses (i) a 10-member Q-ensemble with random 2-Q target subsampling, (ii) high-UTD critic training (20 critic updates per environment step), and (iii) a fixed 50/50 offline-online replay mixture. All of the relevant parameters are reported in Table 6.

### A.3.2 RLPD with Low-Success Synthetic Data vs SAC from Scratch

To analyze whether low-success synthetic datasets can still bootstrap online learning, we select two target tasks with the lowest non-zero synthetic dataset success rates after iterative generation (8% and 12%, measured by TD3-BC offline training). All necessary experiment hyperparameters are listed in Table 7.

## A.4 Computational Cost Analysis

### A.4.1 Computational Requirements

All experiments were conducted on a SLURM-managed GPU cluster equipped with NVIDIA RTX 2080 Ti, RTX 3090, RTX A10, RTX A40, RTX A6000, and L40 GPUs. Training jobs were distributed across these node types, and all model and batch-size configurations were selected to run reliably within the memory constraints of this mixed hardware environment.

Table 5: Compositional RL baseline hyperparameters for multitask TD3-BC training.

| COMPONENT | HC RL (Mendez et al., 2022a) | S+C RL (Ours) |
|---|---|---|
| Algorithm | TD3-BC | TD3-BC |
| Training tasks | 14 | 14 |
| Test tasks | 32 | 32 |
| State dimension | 93 (with task IDs) | 93 (with task IDs) |
| Action dimension | 8 | 8 |
| Actor architecture | Hardcoded compositional MLP | Semantic compositional transformer |
| Actor output dimension | 8 | 8 |
| Critic architecture | Hardcoded compositional MLP (state+action) | Semantic compositional transformer (state+action) |
| Critic output dimension | 1 | 1 |
| Compositional module sizes | $(32), (32, 32), (64, 64, 64), (64, 64, 64)$ | – |
| Compositional hierarchy | Obstacle $\rightarrow$ Object $\rightarrow$ Subtask $\rightarrow$ Robot | – |
| Transformer hidden size | – | 72 |
| Transformer depth | – | 1 |
| Transformer heads | – | 4 |
| Transformer MLP ratio | – | 1.20 |
| Policy network capacity | 107.94K parameters | 106.29K parameters |
| Learning rate | $3 \times 10^{-4}$ | $3 \times 10^{-4}$ |
| Optimizer | Adam | Adam |
| Batch size | 3584 (14 tasks $\times$ 256) | 1792 (14 tasks $\times$ 128) |
| Training steps | 50,000 | 50,000 |
| Discount factor ($\gamma$) | 0.99 | 0.99 |
| Target network update ($\tau$) | 0.005 | 0.005 |
| Regularization coefficient ($\alpha$) | 2.5 | 2.5 |
| Policy noise | 0.2 | 0.2 |
| Noise clip | 0.5 | 0.5 |
| Policy update frequency | 2 | 2 |
| Evaluation frequency | 5,000 steps | 5,000 steps |
| Evaluation episodes | 10 per test task | 10 per test task |
| State normalization | Yes | Yes |
| Reward normalization | No | No |
| Training seeds | 0–14 | 0–14 |

### A.4.2 End-to-End Runtime and GPU Usage of the Iterative Compositional Generation Process

All runtime measurements reported below were obtained on a single NVIDIA RTX A6000 GPU. While experiments were conducted on a SLURM-managed cluster with heterogeneous GPU types, the runtime numbers in Table 8 are standardized to A6000 hardware for fair comparison.

For each task, we generate 1 million environment transitions using the trained diffusion model. The reported data generation time therefore corresponds to generating 1 million transitions for a single task. Offline policy learning time is also reported per task. Since the offline policy learning procedure and dataset size are identical across methods, its runtime remains the same.

Within a task, data generation and policy learning are executed sequentially and therefore cannot be parallelized. However, across different tasks, these jobs can be scheduled concurrently, subject to cluster resource availability.

Table 6: RLPD with real expert data hyperparameters.

| COMPONENT | RLPD (Ball et al., 2023) |
|---|---|
| Offline data source | Real expert demonstrations (14 training tasks) |
| Evaluation tasks | 32 held-out tasks |
| State dimension | 93 (with task IDs) |
| Action dimension | 8 |
| Policy network | MLP, 2 hidden layers (256) |
| Critic network | MLP, 2 hidden layers (256) |
| Critic layer normalization | Yes |
| Number of Q-networks | 10 (ensemble) |
| Target Q subsample size | 2 (from 10) |
| Policy learning rate | $3 \times 10^{-4}$ |
| Q-network learning rate | $3 \times 10^{-4}$ |
| Optimizer | Adam |
| Batch size | 256 |
| Update-to-data ratio (UTD, critic updates) | 20 |
| Offline/online replay mix ratio | 50/50 (fixed) |
| Environment interaction budget | 100,000 steps |
| Discount factor ($\gamma$) | 0.99 |
| Target network update ($\tau$) | 0.005 |
| Entropy coefficient ($\alpha$) | 0.2 (autotuned) |
| Policy update frequency | 1 per environment step |
| Target critic update frequency | 1 per critic update |
| Evaluation frequency | 10,000 steps |
| Evaluation episodes | 10 per task |
| Training seeds | 0–4 |

Let $N_{\text{unsolved}}$ denote the number of remaining tasks in a given iteration and $G$ the number of GPUs available for task-level jobs. The wall-clock time of one iteration can then be approximated as

$$T_{\text{iter}} = T_{\text{diffusion}} + \left\lceil \frac{N_{\text{unsolved}}}{G} \right\rceil (T_{\text{gen}} + T_{\text{policy}}),$$

where $T_{\text{diffusion}}$ is the fixed retraining cost per iteration, and $T_{\text{gen}} + T_{\text{policy}}$ is the sequential per-task cost. In the ideal case where $G \geq N_{\text{unsolved}}$, all tasks are processed in parallel and the iteration time reduces to $T_{\text{diffusion}} + (T_{\text{gen}} + T_{\text{policy}})$. Conversely, if $G = 1$, the iteration cost scales linearly with $N_{\text{unsolved}}$.

In the ideal case where sufficient GPUs are available to process all unsolved tasks concurrently, the per-iteration runtime reduces to $T_{\text{diffusion}} + (T_{\text{gen}} + T_{\text{policy}})$, since data generation and policy learning for different tasks can be executed in parallel. Based on Table 8, this corresponds to approximately 1.49 hours for the Monolithic model and 6.80 hours for S+C DiT.

Under our cluster allocation constraints, however, only partial parallelization across tasks was possible. The effective iteration time therefore depended on the number of available GPUs and the scheduling state of the shared cluster. In practice, the first iteration in the 14/64 experimental setting required approximately 4.30 hours for the Monolithic model and 28.53 hours for the S+C DiT model, averaged over three independent diffusion training seeds. These values reflect typical resource availability during our experiments rather than a strict theoretical bound. In subsequent iterations, as $N_{\text{unsolved}}$ decreases, the amount of per-task computation reduces, leading to correspondingly shorter iteration times.

Table 7: Online RL baselines on low-success target tasks.

| COMPONENT | RLPD (Ball et al., 2023) | Online SAC |
|---|---|---|
| Offline data source | Task-specific synthetic dataset | None |
| Evaluation tasks | 2 rare-success target tasks | 2 rare-success target tasks |
| State dimension | 93 (with task IDs) | 93 (with task IDs) |
| Action dimension | 8 | 8 |
| Policy network | MLP, 2 hidden layers (256) | MLP, 2 hidden layers (256) |
| Critic network | MLP, 2 hidden layers (256) | MLP, 2 hidden layers (256) |
| Critic layer normalization | Yes | Yes |
| Number of Q-networks | 10 (ensemble) | 2 |
| Target Q subsample size | 2 (from 10) | – |
| Policy learning rate | $3 \times 10^{-4}$ | $3 \times 10^{-4}$ |
| Q-network learning rate | $3 \times 10^{-4}$ | $1 \times 10^{-3}$ |
| Optimizer | Adam | Adam |
| Batch size | 256 | 256 |
| Update-to-data ratio (UTD, critic updates) | 20 | 1 |
| Environment interaction budget | 500,000 steps | 5,000,000 steps |
| Discount factor ($\gamma$) | 0.99 | 0.99 |
| Target network update ($\tau$) | 0.005 | 0.005 |
| Entropy coefficient ($\alpha$) | 0.2 (autotuned) | 0.2 (autotuned) |
| Policy update frequency | 1 per environment step | 1 per environment step |
| Target critic update frequency | 1 per critic update | 1 per training step |
| Evaluation frequency | 50,000 steps | 50,000 steps |
| Evaluation episodes | 10 per task | 10 per task |
| Training seeds | 0–4 | 0–4 |

Table 8: Wall-clock runtime measured on a single NVIDIA RTX A6000 GPU. Diffusion training time is reported per model. For each task, 1 million transitions are generated; thus data generation and offline policy learning times are both reported per task. All times are reported in hours.

| COMPONENT | MONOLITHIC (Lu et al., 2023) | S+C DiT (Ours) |
|---|---|---|
| Diffusion Training (per model) | 0.98 | 4.12 |
| Data Generation (1M transitions / task) | 0.23 | 2.40 |
| Offline Policy Learning (per task) | 0.28 | 0.28 |

## A.5 Task List

For the results in Section 4.7.1, we use the ten task lists released by Hussing et al. (2024). For the IIWA-only experiments in Section 4, we construct a train/test split over the full IIWA task space, defined by all combinations of the IIWA robot with:

- **Object**: Box, Dumbbell, Hollowbox, Plate,

- **Obstacle**: GoalWall, None, ObjectDoor, ObjectWall,

- **Objective**: PickPlace, Push, Shelf, Trashcan.

This yields $4 \times 4 \times 4 = 64$ tasks. We generate a random split using seed 0, producing 32 training and 32 test tasks with no overlap. We use the first 14 training tasks for all diffusion and multitask policy experiments, and evaluate on all 32 held-out test tasks. Table 9 lists the tasks, and Figures 14 and 15 visualize the split.

Table 9: Training and test tasks used for IIWA-only experiments.

| Training Tasks (14) | Test Tasks (32) |
| --- | --- |
| 1. IIWA, Box, ObjectDoor, Trashcan | 1. IIWA, Dumbbell, GoalWall, Shelf |
| 2. IIWA, Hollowbox, ObjectDoor, PickPlace | 2. IIWA, Box, None, PickPlace |
| 3. IIWA, Dumbbell, ObjectDoor, PickPlace | 3. IIWA, Box, GoalWall, Shelf |
| 4. IIWA, Dumbbell, ObjectWall, Push | 4. IIWA, Hollowbox, None, PickPlace |
| 5. IIWA, Plate, None, Shelf | 5. IIWA, Dumbbell, ObjectDoor, Push |
| 6. IIWA, Box, GoalWall, Trashcan | 6. IIWA, Box, None, Shelf |
| 7. IIWA, Plate, ObjectWall, Shelf | 7. IIWA, Plate, None, PickPlace |
| 8. IIWA, Hollowbox, GoalWall, Trashcan | 8. IIWA, Dumbbell, None, Shelf |
| 9. IIWA, Box, ObjectWall, Shelf | 9. IIWA, Dumbbell, ObjectDoor, Shelf |
| 10. IIWA, Box, None, Trashcan | 10. IIWA, Hollowbox, GoalWall, PickPlace |
| 11. IIWA, Plate, ObjectWall, PickPlace | 11. IIWA, Dumbbell, GoalWall, Trashcan |
| 12. IIWA, Box, GoalWall, PickPlace | 12. IIWA, Plate, ObjectDoor, Push |
| 13. IIWA, Box, None, Push | 13. IIWA, Plate, ObjectDoor, Shelf |
| 14. IIWA, Box, ObjectDoor, Shelf | 14. IIWA, Hollowbox, None, Trashcan |
| | 15. IIWA, Box, ObjectDoor, PickPlace |
| | 16. IIWA, Box, ObjectDoor, Push |
| | 17. IIWA, Hollowbox, None, Shelf |
| | 18. IIWA, Dumbbell, ObjectWall, Shelf |
| | 19. IIWA, Hollowbox, GoalWall, Shelf |
| | 20. IIWA, Box, ObjectWall, Push |
| | 21. IIWA, Hollowbox, ObjectWall, Shelf |
| | 22. IIWA, Hollowbox, None, Push |
| | 23. IIWA, Plate, GoalWall, Shelf |
| | 24. IIWA, Plate, ObjectDoor, PickPlace |
| | 25. IIWA, Plate, GoalWall, Trashcan |
| | 26. IIWA, Dumbbell, GoalWall, PickPlace |
| | 27. IIWA, Hollowbox, ObjectDoor, Trashcan |
| | 28. IIWA, Dumbbell, ObjectWall, Trashcan |
| | 29. IIWA, Plate, None, Push |
| | 30. IIWA, Plate, GoalWall, Push |
| | 31. IIWA, Dumbbell, None, Push |
| | 32. IIWA, Plate, GoalWall, PickPlace |

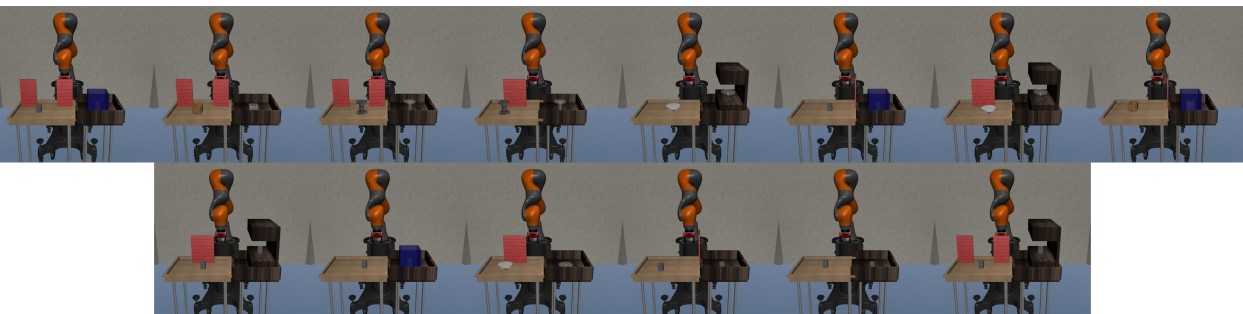

Figure 14: Visualization of the 14 training tasks used in the IIWA-only split. Tasks are shown in numerical order (1–14), arranged left-to-right and top-to-bottom. Each image depicts one unique combination of *Object*, *Obstacle*, and *Objective* paired with the IIWA robot.

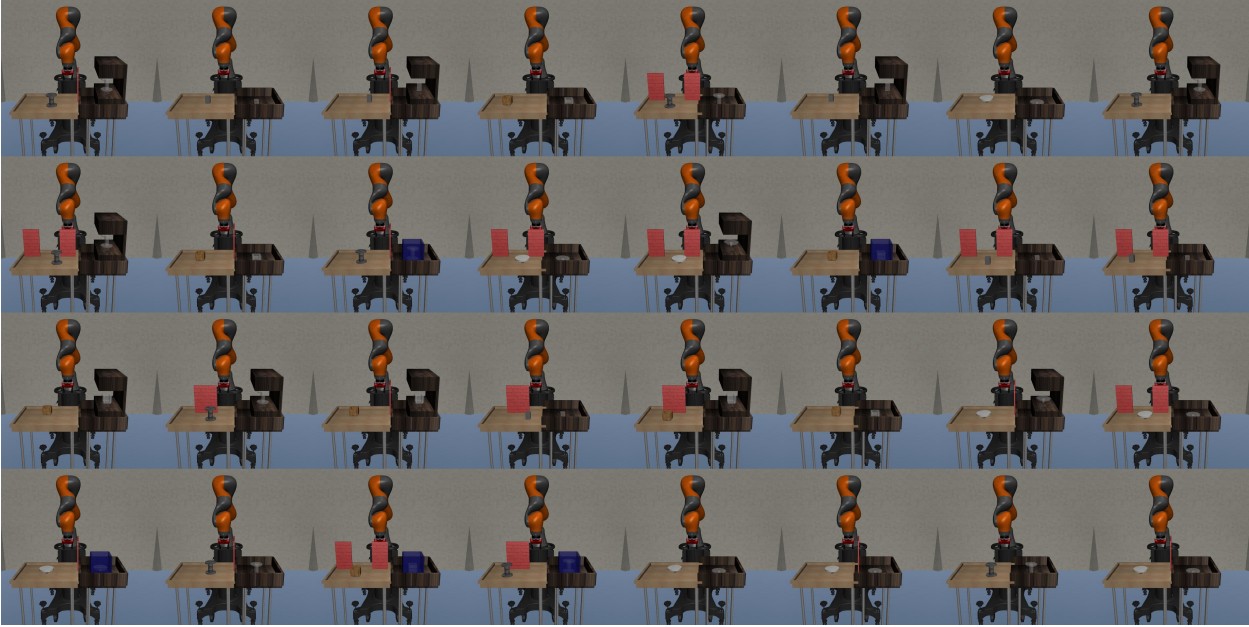

Figure 15: Visualization of the 32 held-out test tasks used for zero-shot evaluation. Tasks are displayed in numerical order (1–32), arranged left-to-right and top-to-bottom. Each image corresponds to a distinct unseen combination of *Object*, *Obstacle*, and *Objective* in the IIWA environment.

