# OpenReview forum: "Iterative Compositional Data Generation for Robot Control"
_TMLR — Accepted by TMLR_

### Review · Reviewer_ERRq · 2026-01-31

**Summary Of Contributions:**

This paper introduces an iterative compositional data generation framework designed to scale robotic learning in multi-object, multi-robot environments. The primary contribution is a Semantic Compositional Diffusion Transformer that factorizes robotic transitions into specific components (robot, object, obstacle, and objective) and learns their interactions via attention. This architecture allows for the zero-shot generation of synthetic training data for unseen task combinations.

Key Strengths:
- The S+C DiT effectively leverages the inherent combinatorial structure of robotic tasks to outperform monolithic models in zero-shot generalization.

Key Weaknesses:
- Validation is restricted to state-based representations in simulation.
- The primary comparison points, such as Synthetic experience replay (NeurIPS2023), while relevant, do not reflect the most recent advances in the rapidly evolving generative robotics field. The reviewer was wondering if there is any more recent state-of-the-art method in this field.
- There are no experiments conducted on physical robotic hardware.

**Additional Comments:**

Addressing the gap between symbolic state simulation and the complexities of real-world robotics remains the most vital step for this work to be truly transformative.

**Audience:**

Yes

**Audience Explanation:**

Audience interested in compositional reinforcement learning, diffusion models for control, and generative data augmentation would find the results helpful.

**Broader Impact Concerns:**

The manuscript does not raise any ethical concerns.

**Claims And Evidence:**

No

**Claims Explanation:**

While the claims regarding the model's performance in the CompoSuite benchmark are well-supported, the evidence for broader applicability is limited. The manuscript argues that this method mitigates the need for expensive physical robot data, yet it relies entirely on symbolic state representations. In real-world scenarios, environments cannot always be neatly factorized into 77-dimensional state vectors. Furthermore, the requirement for online evaluation to filter "good" datasets currently necessitates environment interaction, which contradicts the claim of reducing real-world costs unless a high-fidelity simulator is already available.

**Requested Changes:**

- The authors are encouraged to provide a detailed discussion and experiment on how this factorized approach can be adapted to pixel-based observations or unstructured environments where "state" is not pre-defined.
- The current comparisons center on SynthER (2023). Given the current timeframe, in which the community has shifted to more realistic and practical robot applications, the authors should consider including or discussing comparisons with more recent "World Model" or "Video Diffusion"- based controllers (e.g., RoboDreamer or similar 2024/2025 architectures) that handle compositional tasks.
- The reviewer would be very interested to learn how the proposed method works in real-robot settings.

---

> ### Author Response · Authors · 2026-03-20
>
> Dear Reviewer ERRq:
>
> Thank you for your thoughtful comments and suggestions! We thank the reviewer for recognizing the effectiveness of our compositional design for zero-shot generalization. Here, we respond to the specific questions and comments the reviewer raises. Please let us know if you have lingering questions and whether we can provide any additional clarifications during the discussion period to improve your rating of our paper.
>
> **Requested change 1**
>
> In this work, we focus on state-based settings to isolate and evaluate the utility of compositional representations and self-improvement in a controlled setting. More precisely, our method relies on factorized representations rather than symbolic states per se, and can therefore be naturally extended to pixel-based observations by incorporating structured perception modules (e.g., segmentation or object-centric representations) that map images into factorized latent components. We believe that doing so is not necessarily required to support our claims about compositionality and is out of scope of the present manuscript. A future direction might consider integrating these ideas with RoboDreamer, however at present this is computationally difficult as even the base RoboDreamer requires at least 100 V100 GPUs according to their manuscript.
>
> We have clarified this point in the revised manuscript and expanded the discussion on real-world applicability. In particular, our approach aligns with large-scale simulation settings in robotics, where privileged state information is available during training and policies are later distilled to operate from partial or visual observations for sim-to-real transfer.
>
> **Requested change 2**
>
> We have expanded the related work discussion in the revised manuscript to clarify the distinction between these methods and our setting. Many recent approaches focus on modeling visual dynamics or improving data diversity within a single task or fixed environment, or use generative models as planners. In contrast, our work addresses compositional generalization across tasks, where new tasks arise from novel combinations of components (e.g., robot, object, obstacle, goal).
>
> Our method explicitly models this compositional structure and generates full transitions for unseen task combinations, enabling direct policy learning. In contrast, prior approaches typically focus on within-task modeling or planning, and do not explicitly address cross-task combinatorial generalization or the generation of large-scale datasets for policy learning. This distinction also motivates our choice of SynthER as a monolithic baseline, as it provides a relevant comparison for evaluating generalization to unseen task combinations rather than within-task diversity.
>
> **Requested change 3**
>
> Our method is designed to operate in large-scale simulation settings, where it can efficiently generate high-quality training data across a wide range of task combinations. In practice, this enables scaling expert policy acquisition without requiring additional real-world data collection.
>
> In a typical sim-to-real pipeline, the generated datasets can be used to train or augment policies in simulation, which are then transferred to real robots via standard techniques such as policy distillation or fine-tuning under partial observations. In this way, our approach reduces the need for expensive real-world interaction by shifting data generation and exploration into simulation. We have incorporated these points into the revised manuscripts.
>
> We agree that real-world evaluation is a valuable suggestion. However, evaluating such large-scale compositional settings directly on physical robots is currently impractical. Our setup considers up to hundreds of task combinations (e.g., 256 tasks), whereas most real-robot studies evaluate on around 10–20 tasks. Replicating this setting would require multiple robotic platforms configured across diverse combinations, as well as repeated evaluations across tasks and iterations of the self-improvement loop, resulting in substantial hardware and runtime requirements.
>
> We therefore view large-scale simulation as a necessary step for studying compositional generalization at scale, enabling the development of methods that can later be deployed more efficiently in real-world systems as robotic infrastructure and evaluation capabilities improve.

---

### Review · Reviewer_h18R · 2026-02-08

**Summary Of Contributions:**

The paper introduces a semantic compositional diffusion transformer that generates synthetic training data for unseen robotic manipulation tasks by learning compositional structure from data. The model factorizes transitions into robot-, object-, obstacle-, and objective-specific components and uses self-attention to learn their interactions. An iterative self-improvement procedure generates data for novel task combinations, validates it with offline RL, and incorporates high-quality synthetic data into subsequent training rounds. Experiments on CompoSuite demonstrate superior zero-shot generalization compared to monolithic baselines, and analysis shows that the learned compositional structure differs from hand-engineered priors.

**Audience:**

Yes

**Audience Explanation:**

This work addresses compositional generalization and data efficiency in robot learning—important problems for the TMLR audience. The comparison between learned vs. hand-coded compositional architectures and the iterative self-improvement mechanism offer interesting insights.

**Broader Impact Concerns:**

No significant concerns

**Claims And Evidence:**

No

**Claims Explanation:**

The 2× improvement over hard-coded compositional RL (Mendez et al., 2022a) with a learned structure is well demonstrated across multiple seeds. The intervention analysis (Figure 7) and attention visualization (Figure 8) provide clear evidence that the model learns meaningful compositional decompositions that differ from engineering priors.

However, the paper missed several key points.

- Missing critical baselines: No comparison with recent compositional generation methods (RoboDreamer [Zhou et al., 2024], Dream to Manipulate [Barcellona et al., 2025], Gao et al., 2024) despite citing them—impossible to assess if gains come from compositional structure or just using transformers
- Contradictory efficiency claims: Requires 2000 online rollouts (10 per task × 50 tasks × 4 iterations) for validation, undermining the core claim of reducing real-world data collection without justification of these costs
- Insufficient scale: Main results use only 14/64 tasks; full-scale 56/256 results relegated to appendix with minimal analysis—inadequate for claims about compositional generalization
- No computational cost analysis: Training time, GPU hours, and generation costs unreported—cannot assess practical feasibility vs. collecting real demonstrations
- Modest absolute performance: 55% success rate with ~90% task coverage means most tasks solved only occasionally, limiting practical utility
- Model collapse claim unsupported: Section 3.3 claims compositional structure prevents collapse but provides no empirical evidence (no quality metrics tracked across iterations

**Requested Changes:**

Please address the points raised above. Additionally, if possible, Figure 4 may be improved by breaking it into modules.

---

> ### Author Response · Authors · 2026-03-20
>
> Dear Reviewer h18R,
>
> We appreciate the reviewer for highlighting the strong empirical gains of our approach, as well as recognizing that our analysis provides evidence of meaningful learned compositional structure.  Below, we respond to your questions and comments. Please let us know if any aspects remain unclear, and we would be glad to provide further clarification during the discussion period.
>
> **Point 1: Baselines**
>
> We first clarify that the cited methods differ in scope and setting from our work. In particular, some of the referenced approaches (e.g., Gao et al., 2024) do not focus on generative modeling of RL transitions, but instead address different problem formulations centered on data collection. The remaining methods (e.g., RoboDreamer, Dream-to-Manipulate) primarily focus on modeling visual dynamics, planning, or improving data diversity within a single task or fixed environment. In contrast, our work explicitly targets compositional generalization across tasks, where new tasks arise from novel combinations of components (robot, object, obstacle, goal), and requires generating full transition datasets for unseen task combinations to enable direct policy learning. Additionally, our current formulation operates on structured state representations, allowing us to isolate and study compositionality in a controlled setting. Extending this framework to image-based observations is an important direction for future work.
>
> To address the reviewer’s concern regarding whether the gains stem from compositional structure or simply the use of transformers, we note that this distinction is explicitly studied in our experimental design, and we have further clarified this in the revised manuscript.
>
> Specifically, we compare three classes of generative architectures under the same training and evaluation pipeline: (1) a monolithic diffusion model following SynthER (Lu et al., 2023), (2) a standard diffusion transformer (Peebles & Xie, 2023) without compositional structure, and (3) variants with increasing levels of structure, including semantic tokenization and our full compositional architecture. Importantly, the standard DiT performs worse than the monolithic baseline, indicating that improvements do not arise from simply replacing the architecture with a transformer. Introducing semantic tokenization yields modest gains, while the full compositional architecture provides the largest improvement.
>
> This layered comparison isolates the effect of compositional structure from architectural choices, and demonstrates that the observed gains are driven by explicitly modeling task factorization rather than by the use of transformers alone.
>
> We have revised the manuscript to emphasize this distinction more clearly.
>
> **Point 2: Efficiency claims**
>
> We thank the reviewer for raising this important point.
>
> The online rollouts in our approach are used only for evaluation and filtering of generated datasets, not for policy learning. Each evaluation consists of 10 rollouts of length 500, corresponding to approximately 5,000 environment steps per task per iteration, and at most 20,000 interactions across four iterations. These interactions are used solely to assess dataset quality and are not used to update the policy or generative model. In future work, it may be possible to replace this evaluation procedure with learned or offline proxies that estimate dataset quality without requiring rollouts. In contrast, learning effective policies for new tasks inherently requires interaction or high-quality data, and cannot be substituted by such proxies.
>
> We also agree that it would be fair to compare to approaches that actually attempt to learn with the same amount of data. We have added an explicit analysis of interaction efficiency in Section 4.5, where we included an empirical comparison to RLPD (Ball et al., 2023), which performs online RL with up to 100,000 environment interactions per task while leveraging the same offline expert dataset from the 14 training tasks. There, we show that our method achieves over 55% success on 32 unseen tasks using at most 20,000 environment interactions, while RLPD fails to achieve meaningful success even after 100,000 interactions.
>
> At the same time, our method generates large synthetic datasets, approximately 1 million transitions per task in our current setting, without additional environment interaction. This shifts the burden from collecting data through interaction to validating generated data, which is substantially more efficient.
>
> We have clarified these points in the revised manuscript.

---

> ### Author Response · Authors · 2026-03-20
>
> **Point 3: Scale to more compositional axes**
>
> We have revised the manuscript to move the full-scale 56/256 results from the appendix to the main paper (Section 4.7) and provide additional analysis. These results show consistent qualitative behavior with the main experiments, supporting the generality of our findings.
>
> In the original submission, these results were included in the appendix to keep the main paper concise and focused within the TMLR page limits. Following the reviewer’s suggestion, we have now incorporated them into the main text for improved clarity and completeness.
>
> Importantly, we clarify that the 14/64 setting used in the main experiments is in fact more challenging from a compositional perspective. While both settings use a similar proportion of training tasks, the 14/64 regime contains fewer observed task combinations in absolute terms, resulting in greater combinatorial sparsity. This makes generalization to unseen task combinations more difficult and amplifies the importance of modeling compositional structure Empirically, this is reflected in our results:the monolithic baseline performs significantly better in the 56/256 setting than in 14/64, while our method remains stable across both regimes.
>
> As a result, the stronger gains observed in the 14/64 setting reflect a more demanding compositional generalization scenario, rather than a limitation in scale. The 56/256 results confirm that our method continues to perform consistently as the number of tasks increases.
>
> We have clarified these points in the revised manuscript.
>
> **Point 4: Computational cost analysis**
>
> We have added a detailed computational cost analysis in the revised manuscript (Appendix B4), where we report training time, GPU usage, and generation costs of the proposed pipeline. This analysis quantifies the overhead of iterative compositional data generation at scale and provides a clearer assessment of its practical feasibility.
>
> While our method introduces additional computation for data generation, this cost is offset by the ability to generate large-scale synthetic datasets without requiring additional data collection. In particular, synthetic data generation can also be parallelized and run without supervision.
>
> We have included these details to better contextualize the computational practicality of our approach.
>
>  **Point 5: Generated data performance and practical utility of ~90% task coverage**
>
> We agree that the previously reported metric reflects the fraction of tasks that are solved at least once over time, which can include occasional successes.
>
> To address this limitation, we have added a new metric in the revised manuscript that measures the coverage of high-quality datasets, defined as the fraction of tasks for which original training tasks and newly generated datasets pass a quality threshold and are incorporated into training.
>
> As shown in Fig. 6, while the fraction of tasks solved at least once reaches around 90%, the coverage of high-quality datasets provides a stricter measure of consistency. In particular, our method increases coverage from roughly 20% to approximately 55%, indicating that it generates reliable and usable datasets across a large portion of the task space, rather than only achieving occasional successes. In comparison, the monolithic baseline reaches only around 35% coverage.
>
> In addition, we include an analysis in Section 4.6 showing that even datasets with low but non-zero success rates can significantly accelerate downstream RL, while learning from scratch without such data fails to achieve any success even after substantially more interaction. This demonstrates that even infrequent successful trajectories can provide meaningful learning signals and significantly accelerate downstream learning The result directly addresses the concern regarding practical utility: the high fraction of tasks solved at least once (~90%) reflects broad coverage of useful learning signals, even when individual successes are infrequent..
>
> Together, these results clarify that the reported performance reflects both broad task coverage and the consistent generation of useful datasets, addressing the concern that performance is driven only by occasional successes. Furthermore, they also demonstrated that even very low-success datasets, which may only contain rare successful trajectories, can still provide practical utility for downstream learning.

---

> ### Author Response · Authors · 2026-03-21
>
> **Point 6: Model collapse**
>
> In the revised manuscript, we have removed the claim regarding prevention of model collapse and clarified the role of the compositional structure more precisely.
>
> Specifically, our architecture localizes updates to task-specific encoder-decoder modules, so that data generated for a particular task only affects the corresponding components, rather than propagating globally across all tasks. This design limits the extent to which low-quality data can affect representations used by other tasks.
>
> While we do not explicitly measure model collapse, we provide empirical evidence that the compositional model maintains stable and consistently improving performance across iterations, while achieving higher success rates and greater coverage of high-quality datasets compared to baseline architectures (Section 4.3, Fig. 6).
>
> We have revised Section 3.3 accordingly to reflect this more precise interpretation.
>
> **Additionally, if possible, Figure 4 may be improved by breaking it into modules.**
>
> We intentionally present Figure 4 at the component level to make the factorized structure of the model explicit, as this is central to our contribution. In particular, showing the individual encoder–decoder pairs and their interactions through the diffusion transformer highlights how compositional structure is represented and learned.
>
> While a more abstract modular diagram could simplify the presentation, it would obscure these component-level interactions, which we believe are important for understanding the model design.

---

### Review · Reviewer_pbza · 2026-03-09

**Summary Of Contributions:**

The paper models robotic manipulation transitions with a semantic compositional diffusion transformer that factorizes a transition into factor-specific tokens for task elements and learns dependencies between them with self-attention. The generator is trained on 14 IIWA-only CompoSuite tasks, then used to synthesize 1M transitions for unseen task combinations. Task-specific TD3-BC policies are trained on these synthetic datasets, and a bootstrapping loop adds back synthetic data for tasks whose learned policies clear a success threshold. On the reported 32 held-out test tasks, the semantic compositional DiT is the strongest diffusion-based method both before and after bootstrapping, reaching about 55% average success after four iterations.

**Audience:**

Yes

**Audience Explanation:**

The paper studies an important question: whether a structured generative model of transitions can improve zero-shot generalization when only a small fraction of compositional tasks is observed during training. I believe this work will be of interest to researchers working on offline RL, diffusion models for control, and structured representations for robot learning.

**Broader Impact Concerns:**

No. I do not find any broader impact concerns arising from this work.

**Claims And Evidence:**

Yes

**Claims Explanation:**

The main empirical claims are supported. Figure 5 shows a clear gap between the semantic compositional DiT and the monolithic, standard DiT, and semantic-only DiT baselines, so the claim that the proposed factorization improves zero-shot generative transfer is justified. The semantic-only ablation is important and well-thought out. The final iterative result is also supported in the sense that the proposed model remains the strongest diffusion-based approach after four rounds of self-improvement. That said, the paper is less convincing on some narrower mechanistic claims:
- Fig. 6 reports best-so-far success and tasks solved at least once, both of which are non-decreasing by construction, so those plots are weaker evidence of stable iterative improvement than raw per-iteration performance would be.
- Likewise, the claim that the architecture partially guards against model collapse is not directly validated, and the attention analysis is based on a separately trained 1-layer transformer rather than the full 8-layer model used for the main results.

**Requested Changes:**

**Critical Changes:**
- Fig. 6 should report raw per-iteration mean success in addition to best-so-far success and tasks solved at least once. As currently defined, both reported metrics are structurally monotonic, so the plots do not by themselves establish that self-improvement is stable rather than accumulative.
- The structural analysis in Sec. 4.4 should be calibrated more carefully. The attention analysis is performed on a separate 1-layer transformer and only before self-improvement, so it does not yet show that the full model learns and preserves the claimed dependency structure throughout the actual iterative training pipeline.
- The paper should quantify the online cost of the iterative filtering loop and relate it to the stated goal of reducing expensive data collection. In the current setup, Alg. 1 still relies on online evaluation of held-out tasks in every round to decide which synthetic datasets are added back into training. Since this online evaluation is central to the method, the paper should state this cost clearly and discuss how it compares with collecting additional demonstrations directly.

**Additional changes:**
- The discussion of model collapse is currently more conjectural than demonstrated. While the model uses factor-specific encoder–decoder pairs, the shared DiT backbone is still updated by all admitted synthetic datasets. A simple retention analysis on the original training tasks across iterations would test whether those shared updates remain stable in practice, or else the claim that the architecture partially guards against collapse should be softened.

---

> ### Author Response · Authors · 2026-03-20
>
> Dear Reviewer pbza,
>
> We sincerely appreciate your insightful and thorough feedback, as well as your recognition of our empirical results and compositional design. Your suggestions were particularly valuable and helped us strengthen both the clarity and rigor of our experiments. Below, we respond to your questions and comments, and we are happy to clarify any remaining points during the discussion period.
>
> **Requested Change 1 (Critical)**
>
> We thank the reviewer for this important suggestion.
>
> To address this concern, we have added the raw per-iteration mean success rate to Fig. 6 (top middle). Unlike the best-so-far success and tasks-solved metrics, which are structurally monotonic, this metric reflects the actual performance achieved at each iteration and is not monotonic by construction.
>
> As shown in Fig. 6, the mean success rate increases consistently across iterations for all architectures, with our compositional approach achieving the largest gains. Importantly, this trend closely mirrors the improvements observed in the best-so-far success metric, indicating that performance improvements are not driven by accumulation effects, but instead reflect genuine improvements in the quality of generated data over iterations.
>
> We have added this analysis and corresponding discussion in the revised manuscript to clarify that the observed gains reflect stable self-improvement.
>
> **Requested Change 2 (Critical)**
>
> In the original version, we presented a single-layer analysis to provide a more direct view of how attention operates between input components. This is particularly important as the transformer beyond a single layer may learn relationships that are difficult to interpret from attention maps alone. Nevertheless, we have extended the analysis to the full model as requested
>
> We provide this additional analysis of the full 8-layer diffusion transformer in Section 4.4 . The updated results are shown in Fig. 9.
>
> Our analysis shows that the learned dependency structure is consistent not only in a single-layer setting, but throughout the full model depth. In particular, we observe a stable diagonal attention pattern across all layers, indicating that each component continues to attend primarily to its corresponding encoder.
>
> Importantly, this structure is preserved across all self-improvement iterations. The relative importance ordering between components remains consistent, with the robot receiving the strongest attention, followed by the goal, while the object and obstacle receive weaker attention. We also observe that attention patterns become more pronounced in intermediate layers, suggesting that compositional interactions are captured within the transformer hierarchy.
>
> These results demonstrate that the model not only learns the intended compositional structure, but also maintains and refines it throughout the iterative training process. We have updated the manuscript accordingly to reflect this more comprehensive analysis.

---

> > ### Author Response · Authors · 2026-03-21
> >
> > **Requested Change 3 (Critical)**
> >
> > We thank the reviewer for highlighting the importance of quantifying and contextualizing the online cost of the iterative filtering loop.
> >
> > In the revised manuscript (Section 4.5), we explicitly quantify the environment interaction required by our method. Each evaluation consists of 10 rollouts of length 500 (≈5,000 transitions) per task per iteration, amounting to at most 20,000 environment interactions per task over four iterations. Importantly, these interactions are used solely to evaluate policy performance and determine whether a generated dataset should be incorporated into training; they are not used for policy or model updates. Notably, this corresponds to at most only 40 trajectories per task in total, which is small compared to typical reinforcement learning approaches that require thousands of trajectories per task.
> >
> > It may be possible to replace this evaluation procedure with learned or offline proxies that estimate dataset quality without requiring rollouts in future work. In contrast, learning effective policies for new tasks inherently requires interaction or high-quality data, and cannot be substituted by such proxies.
> >
> > We further clarify how this cost compares to collecting additional demonstrations. While our approach does require environment interaction, the evaluation rollouts are short, require no expert supervision, and can be executed autonomously and in parallel. In contrast, collecting demonstrations would require either expert policies or human teleoperation for each new task, making it significantly more expensive and less scalable. Thus, our method replaces costly data collection with relatively inexpensive evaluation.
> >
> > Finally, we complement this analysis with an empirical comparison to RLPD (Ball et al., 2023), which performs online RL with up to 100,000 environment interactions per task while leveraging the same offline expert dataset from the 14 training tasks. Despite this substantially larger interaction budget and direct training on the target task, RLPD achieves near-zero success, whereas our approach reaches over 55% success using only ~20,000 interactions. This demonstrates that a small amount of evaluation interaction suffices to validate large amounts of synthetic data and significantly reduce the interaction required compared to directly learning policies on new tasks through environment interaction.
> >
> > **Requested Change 4 (Additional)**
> >
> > In the revised manuscript, we have removed the claim regarding prevention of model collapse and clarified the role of the compositional structure more precisely.
> >
> > Specifically, our architecture localizes updates to task-specific encoder–decoder modules, so that data generated for a particular task primarily affects the corresponding components rather than propagating uniformly across all tasks. This design limits the extent to which low-quality data from one task can impact representations used by others, but does not eliminate the possibility of degradation in the shared backbone.
> >
> > We do not explicitly measure model collapse in this work. Instead, we position this as a structural property of the architecture and avoid making claims about preventing collapse. We have revised Section 3.3 accordingly to reflect this more precise interpretation.

---

### Author Response · Authors · 2026-03-20
**Updated Manuscript (revision 1) and Paper Summary**

Dear Reviewers and Area Chair,

We thank you for your insightful feedback. We have revised the manuscript accordingly and incorporated additional analyses and clarifications. We summarize our key changes here and provide individual responses to answer specific questions raised by the reviewers (highlighting additions in blue and deletions in red in the revised manuscript):

1. Stability of self-improvement (Figure 6 and 13, top middle): We report the raw per-iteration mean success rate in Fig. 6 (top middle) and Fig. 13 for the scaling setting. This metric shows that improvements are consistent across iterations and are not driven by best-of or accumulated effects.

2. Task coverage and data quality (Figure 6 and 13, bottom middle): We additionally report task coverage (number of high-quality datasets) in Fig. 6 and Fig.13 (bottom middle). Coverage measures the fraction of tasks for which synthetic datasets pass the quality threshold and are incorporated into training. This demonstrates that performance is not driven by occasional successes on many tasks, but by the consistent generation of reliable datasets that are useful for downstream learning.

3. Attention analysis (Section 4.4): We extend Fig. 9 to the full 8-layer model across iterations. The learned dependency structure remains stable throughout training, with a consistent importance ordering (robot > goal > object > obstacle), indicating preserved and progressively refined compositional relationships.

4. Environment interaction efficiency (Section 4.5): To directly address the cost of environment interaction raised by the reviewers, we quantify the number of interactions required by our iterative filtering procedure and compare it to RLPD (Ball et al., 2023), a standard offline to online RL baseline. Compared to RLPD, which uses all 14 training datasets as offline data as well as online interactions from the test tasks, our method solves tasks within 20k environment steps, while RLPD fails completely even after 100k environment steps per task, demonstrating substantially improved interaction efficiency.

5. Utility of low-success datasets (Section 4.6): We show that datasets containing even rare successful trajectories significantly accelerate RL. In contrast, online RL without such data fails to achieve any success even after an order of magnitude more interaction steps, highlighting the importance of broad task coverage.

6. Full-scale experiments (Section 4.7): We move the 56/256 results to the main paper and clarify that both the monolithic baseline and our semantic compositional architecture achieve higher initial zero-shot performance in this setting due to increased task diversity, and exhibit similar coverage across iterations. The semantic compositional model nevertheless achieves consistently higher success due to better data quality, demonstrating that our approach scales robustly to larger task sets. Note that the 14/64 setting is more challenging due to increased combinatorial sparsity, leading to a larger performance gap between methods.

7. Related work clarification (Section 5): We expand comparisons to prior world modeling and video generation approaches, emphasizing that our method generates full transitions for policy learning on unseen tasks, rather than focusing on data collection, visual diversity within single tasks, or planner-based usage.

8. Model collapse discussion Section 3.3): We adjusted the claim that the architecture prevents model collapse by communicating more clearly that the modular architecture localizes task-specific updates, which mitigates the propagation of low-quality data, rather than fully preventing model collapse.

9. Real-world relevance (Section 6): We expand discussion of applicability to robotics, highlighting advantages in simulation settings and the potential to scale sim-to-real transfer by accelerating expert policy acquisition for downstream distillation. We also outline extensions to visual observations which we believe are out of scope for the present manuscript.

10. Compute cost analysis (Appendix B4): We add a detailed large-scale cost analysis.

Please let us know if there are any remaining questions, as we would be happy to provide further clarification, and we hope these revisions help clarify our contributions and improve the assessment of the paper.

---

### Decision · Action_Editor_ejL6 · 2026-04-10

**Recommendation:** Accept with minor revision

**Additional Comments:**

I recommend accept with minor revision, where the minor revision consists of the suggestions for minor rephrasings explained in my decision above.

**Audience:**

Yes

**Audience Explanation:**

Robotics, RL, ML, clearly in scope.

**Claims And Evidence:**

Yes

**Claims Explanation:**

Reviewers and I, for the most part, agree that claims made are adequately supported. The only point of potential discussion here is related to the stated motivation behind the paper involving that we want to avoid having to collect much real-world data, but there is no empirical evaluation of trained robots (trained on the synthetically generated data) in the real world.

I do not view this as a major concern, but do think the paper could benefit from perhaps very slight adaptations to how contributions are phrased: I'm looking primarily at the Conclusion here, with sentences such as "This can ultimately reduce data collection and engineering costs for real-world robotic systems [...]". It is fine to claim these as hypothesised benefits, but perhaps they are not 100% certain yet. Some acknowledgement of this limitation of the work / future work on evaluating on real-world tasks (which could just be a subset/proof of concept of all tasks) would not be out of place.

---

> ### Author Response · Authors · 2026-05-08
>
> We sincerely thank the Action Editor and reviewers for their thoughtful and constructive feedback throughout the review process. In the revised version, we adapted the wording of several statements regarding reductions in real-world data collection and engineering costs to better reflect the current scope of the paper. We also added a new Limitations and Future Work section explicitly acknowledging that the current evaluation focuses on large-scale simulated compositional settings, while discussing real-world robotic evaluation as an important direction for future work.